# Molecular mechanism of Oxr1p mediated disassembly of yeast V-ATPase

Md. Murad Khan [ID] & Stephan Wilkens [ID] [✉]

## Abstract

The eukaryotic vacuolar H⁺-ATPase (V-ATPase) is regulated by reversible disassembly into autoinhibited $V_1$-ATPase and $V_o$ proton channel subcomplexes. We recently reported that the TLDc protein Oxr1p induces V-ATPase disassembly in vitro. Whether and how Oxr1p is involved in enzyme disassembly in vivo, however, is not known. Here, using yeast genetics and fluorescence microscopy, we show that Oxr1p is essential for efficient V-ATPase disassembly in the cell. Supporting biochemical and biophysical in vitro experiments show that whereas Oxr1p-driven holoenzyme disassembly can occur in the absence of nucleotides, the presence of ATP greatly accelerates the process. ATP hydrolysis is needed, however, for subsequent release of Oxr1p so that the free $V_1$ can adopt the autoinhibited conformation. Overall, our study unravels the molecular mechanism of Oxr1p-induced disassembly that occurs in vivo as part of the canonical V-ATPase regulation by reversible disassembly.

**Keywords** Vacuolar H⁺-ATPase; Oxr1p; TLDc Domain; Reversible Disassembly
**Subject Category** Membranes & Trafficking

## Introduction

Vacuolar H⁺-ATPases (V-ATPases; $V_1V_o$-ATPases) are ATP hydrolysis-driven proton pumps that acidify the lumen of intracellular organelles, a process essential for pH and ion homeostasis, protein sorting and degradation, endocytosis, cell signaling and neurotransmitter loading and release (Collins and Forgac, 2020; Eaton et al, 2021b; Futai et al, 2019; Kane, 2016; Maxson and Grinstein, 2014). In higher organisms, V-ATPase can also localize to the plasma membrane of polarized cells, where enzyme function is required for bone resorption, sperm maturation, and urine acidification (Breton and Brown, 2013). V-ATPase has been implicated in a broad spectrum of clinical conditions, including renal tubular acidosis, deafness, osteopetrosis, cutis laxa, diabetes, neurodegeneration, male infertility, and cancer (Bagh et al, 2017; Brown et al, 1997; Frattini et al, 2000; Karet, 1999;

Sun-Wada et al, 2006; Van Damme et al, 2017). The enzyme also plays an important role in pathogen entry and viral replication, making V-ATPase an important drug target (Gordon et al, 2020; Lindstrom et al, 2018; Marjuki et al, 2011; Santos-Pereira et al, 2021; Sreelatha et al, 2015; Xu et al, 2010).

Yeast V-ATPase is a rotary motor enzyme consisting of 15 different core subunits that are organized into two subcomplexes: a cytosolic $V_1$-ATPase composed of $A_3B_3CDE_3FG_3H$, and a membrane-embedded $V_o$ proton channel containing $ac_8c'c''def$ (Fig. 1A, left panel) (Harrison and Muench, 2018; Vasanthakumar and Rubinstein, 2020; Wilkens et al, 2021). Cyclic ATP hydrolysis on $V_1$'s three catalytic sites located at alternating A:B interfaces drives rotation of a central rotor made of $V_1$ subunits D and F, which is coupled to $V_o$'s ring of *c* subunits (*c*-ring) through subunit *d*. The rotating *c*-ring carries protons between cytosolic and luminal facing aqueous half-channels found in the membrane integral C-terminal domain of $V_o$ subunit *a* ($a_{CT}$). Three heterodimers of subunits E and G (referred to as EG1-3) function as peripheral stator stalks that link $V_1$'s $A_3B_3CH$ with $V_o$ *a*'s N-terminal domain ($a_{NT}$) to keep the catalytic core static relative to the proton channel during rotary catalysis.

V-ATPase activity is tightly regulated, most prominently by a process referred to as *reversible disassembly* (Kane, 1995; Sumner et al, 1995). Under glucose-fed conditions in yeast, for example, V-ATPases are found in an active, assembled state on vacuolar membranes (Parra and Kane, 1998). However, upon glucose starvation, $V_1$ and subunit C detach from $V_o$, with concomitant *silencing* (or *autoinhibition*) of the activities of the individual $V_1$ and $V_o$ subcomplexes (Couoh-Cardel et al, 2015; Parra et al, 2000) (Fig. 1A, right panel). The process is fully reversible in that $V_1$ and C reassemble with $V_o$ to restore holoenzyme activity when glucose is restored. Reversible disassembly has been extensively studied at the cellular level, and it is known that the disassembly step requires active V-ATPase and intact microtubules, whereas (re)assembly depends on a heterotrimeric chaperone called *RAVE* (**R**egulator of the H⁺-**A**TPase of **V**acuolar and **E**ndosomal membranes), as well as the glycolytic enzymes aldolase and phosphofructokinase (Chan and Parra, 2014; Lu et al, 2007; Parra and Kane, 1998; Seol et al, 2001; Smardon and Kane, 2007; Xu and Forgac, 2001). However, the molecular mechanism of the process at the level of the enzyme is poorly understood. Reversible disassembly, initially described for the enzymes from yeast and insects, is now also recognized as a regulatory mechanism for the mammalian V-ATPase (Bodzeta et al, 2017; Collins and Forgac, 2020; Ratto et al, 2022).

Department of Biochemistry and Molecular Biology, SUNY Upstate Medical University, Syracuse, NY 13210, USA. ✉E-mail: wilkenss@upstate.edu

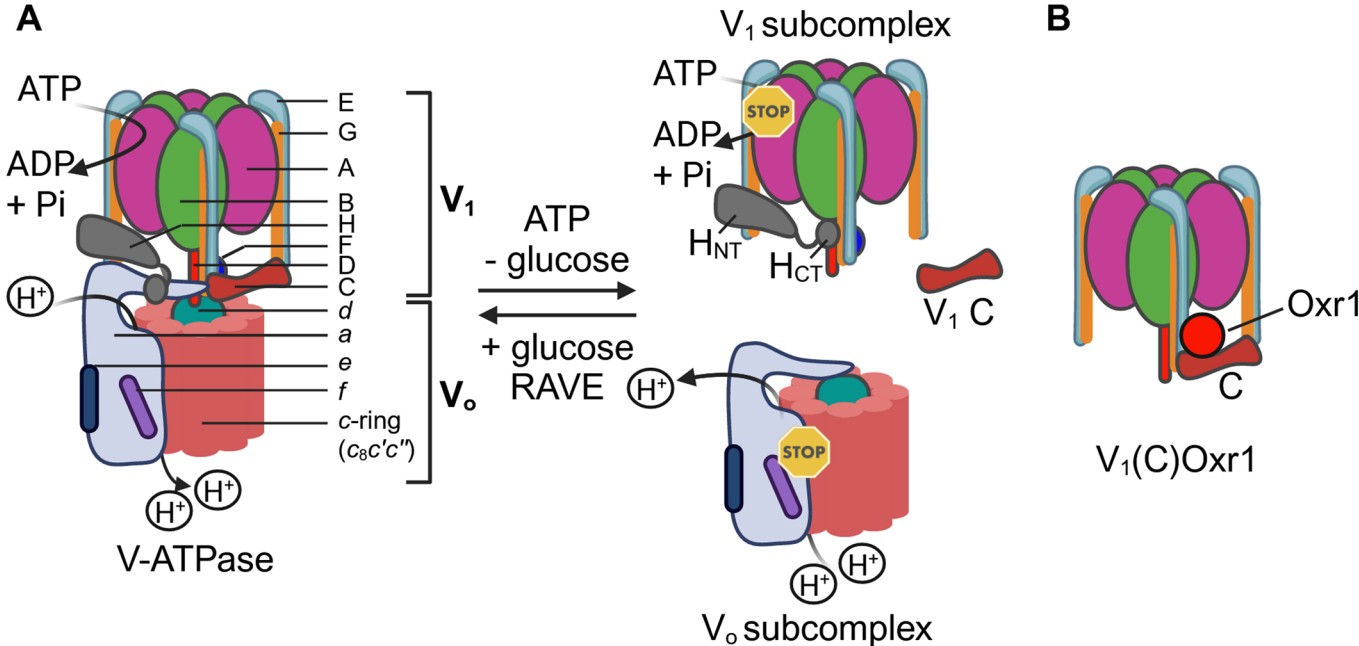

**Figure 1. Structure and regulation of yeast V-ATPase and the $V_1$(C)Oxr1p complex.**

(A) V-ATPase is an ATP hydrolysis-driven proton pump consisting of two subcomplexes, a cytosolic $V_1$-ATPase and a membrane-bound $V_o$ proton channel. The $V_1$ contains eight subunits (designated A to H), and the $V_o$ seven (*a, c, c', c'', d, e, f*), with some of these subunits present in multiple copies. The activity of V-ATPase is regulated by *reversible disassembly*. In yeast, upon glucose withdrawal, $V_1$ and the C subunit dissociate from $V_o$, with both $V_1$ and $V_o$ subcomplexes becoming autoinhibited. Efficient reassembly requires an assembly chaperone (RAVE complex). (B) Schematic of the Oxr1p-bound $V_1$ subcomplex ($V_1$(C)Oxr1p). Labeling as in (A). The figure was created with BioRender.com.

We recently obtained a cryoEM structure of a mutant $V_1$ subcomplex bound to subunit C and **Ox**idation **R**esistance protein 1 (Oxr1p), a cellular factor not seen associated with V-ATPase previously (Khan et al, 2022) (Fig. 1B). Yeast Oxr1p is a member of the family of *TLDc* (**T**re2/Bub2/Cdc16 (TBC), lysin motif (**L**ysM), **D**omain **c**atalytic) domain proteins, with the TLDc domain highly conserved from yeast to human (Finelli and Oliver, 2017). Subsequent in vitro experiments showed that recombinant Oxr1p acted as a V-ATPase inhibitor by inducing enzyme disassembly (Khan et al, 2022). However, whether Oxr1p-induced enzyme disassembly observed in vitro plays a role in canonical V-ATPase regulation by reversible disassembly in vivo, or a novel mechanism of V-ATPase quality control, is not known.

Here, we have generated *OXR1* deletion and overexpression yeast strains to study the physiological role of Oxr1p in V-ATPase regulation. We find that neither deletion nor overexpression of *OXR1* results in a V-ATPase loss-of-function (Vma−) phenotype. However, whereas *OXR1* overexpression leads to fewer $V_1$-ATPases on vacuoles compared to wild-type, *OXR1* deletion vacuoles have more. Using fluorescence microscopy, we find that V-ATPases in the *OXR1* deletion strain remain assembled upon glucose withdrawal, a condition that causes rapid disassembly of the enzyme in wild-type cells. We further find that Oxr1p initially remains bound to $V_1$ following V-ATPase disassembly, and that release of Oxr1p from $V_1$ is ATP-dependent and requires the presence of the inhibitory H subunit. Taken together, the results of our experiments provide a detailed mechanism of regulated V-ATPase disassembly and Oxr1p's role in the process.

## Results

### Oxr1p is capable of disassembling inactive V-ATPase

Prior studies in yeast had shown that glucose withdrawal-induced V-ATPase disassembly requires catalytically active enzymes (Parra and Kane, 1998). On the other hand, more recent data in higher organisms suggested that loss of TLDc domain protein function negatively impacts V-ATPase activity (Merkulova et al, 2018). As these two observations appeared to be at odds with our finding that Oxr1p disassembled V-ATPase in vitro without the need for ATP, we speculated that Oxr1p may serve in quality control by selectively removing $V_1$ subcomplexes that were rendered inactive in vivo due to e.g., oxidative damage (Khan et al, 2022). At the onset of the current study, we therefore tested Oxr1p's ability to release $V_1$ from vacuoles harboring inactive V-ATPases. For this, we purified vacuoles from a yeast strain deleted for the *CYS4* gene, which has been shown to reduce V-ATPase activity by more than 50% (Oluwatosin and Kane, 1997) (Appendix Fig. S1A). As an alternative approach, we treated wild-type vacuoles with hydrogen peroxide ($H_2O_2$), which is known to inhibit V-ATPases due to oxidation of active site cysteine residues (Liu et al, 1997), and the V-ATPase-specific inhibitor, Concanamycin A (ConA) (Appendix Fig. S1A). However, when we tested Oxr1p's impact under these conditions, we found that Oxr1p was equally efficient in releasing $V_1$ from vacuoles with inactive V-ATPases compared to wild-type vacuoles (Appendix Fig. S1B-D). Overall, this shows that whereas Oxr1p is able to release $V_1$ from inactive V-ATPases, it appears that

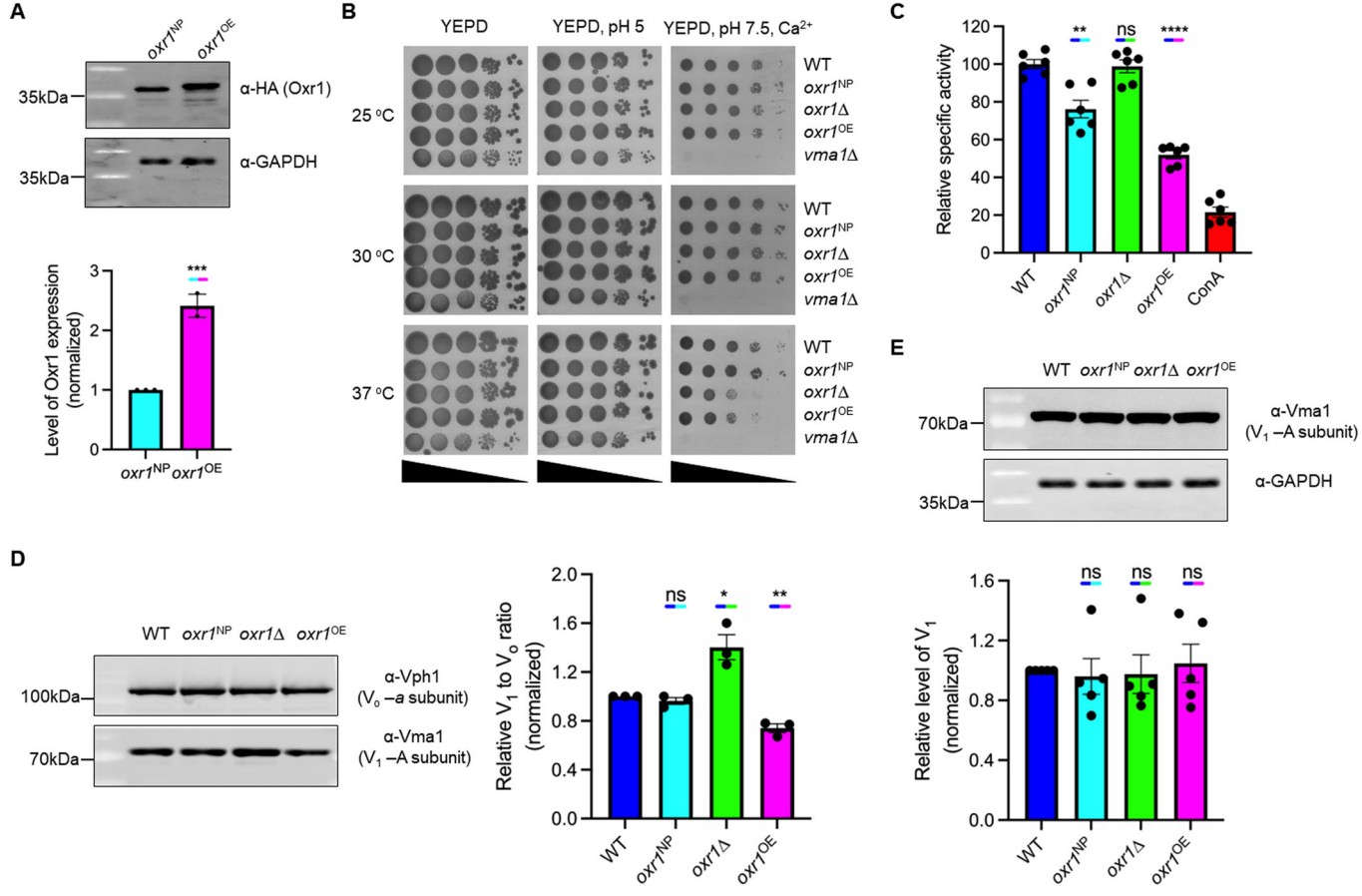

**Figure 2. *OXR1* overexpression lowers V-ATPase activity by reducing the amount of $V_1$ on vacuoles.**

(**A**) *OXR1* was deleted from wild-type yeast, and the resulting *OXR1* deletion strain (*oxr1Δ*) was transformed with plasmids for expression of C-terminally HA-tagged Oxr1p under the native promoter (*oxr1*$^{NP}$) or via a high-copy number YEp352 plasmid (*oxr1*$^{OE}$). Upper panel, immunoblot of equal amounts of total lysates from exponentially growing *oxr1*$^{NP}$ and *oxr1*$^{OE}$ cells probed with α-HA and α-GAPDH (loading control) antibodies. A representative blot from three independent experiments is shown. Lower panel, averages of relative band intensities from three experiments. Data are presented as mean ± SEM. (**B**) Growth phenotypes of *OXR1* deletion and overexpression yeast strains. Exponentially growing wild-type (WT), *oxr1*$^{NP}$, *oxr1Δ*, *oxr1*$^{OE}$, and *vma1Δ* (V-ATPase catalytic A subunit deletion) strains were serially tenfold diluted and spotted on YEPD, YEPD pH 5 and YEPD pH 7.5 + 60 mM $CaCl_2$ plates and incubated at the indicated temperatures for 3–4 days. A representative assay from a total of three repeats is shown. (**C**) ATPase activities of purified vacuoles from *OXR1* mutant strains. Vacuoles from the WT, *oxr1*$^{NP}$, *oxr1Δ*, and *oxr1*$^{OE}$ strains were purified in parallel, and ATPase activities were measured immediately thereafter. Relative activities of vacuoles were normalized against WT activity. Specific ATPase activities of WT vacuoles were between 3.5 and 4.5 μmol × (min × mg)$^{-1}$. Concanamycin A (ConA) is a specific inhibitor of V-ATPase and used as an additional control. Individual data points of six measurements from three biological preparations are shown. Data are presented as mean ± SEM. (**D**) Relative ratios of $V_1$ to $V_o$ on vacuoles purified from *OXR1* mutant strains. Left panel, equal amounts of purified vacuoles from WT, *oxr1*$^{NP}$, *oxr1Δ*, and *oxr1*$^{OE}$ strains were analyzed by western blot using α-Vma1p ($V_1$ subunit A; 8B1F3) and α-Vph1p ($V_o$ subunit a; 10D7) antibodies. Band intensities were quantified, and relative ratios of $V_1$ to $V_o$ were normalized against WT. A representative of three blots from three biological preparations is shown. Right panel, averages of relative band intensities from three experiments. Data are presented as mean ± SEM. (**E**) Relative amounts of $V_1$ in total cell lysates from *OXR1* mutant strains. Top panel, total cell lysates prepared as described in "Methods" from WT, *oxr1*$^{NP}$, *oxr1Δ* and *oxr1*$^{OE}$ strains were separated by SDS-PAGE and analyzed by western blot using α-Vma1p ($V_1$ subunit A) and α-GAPDH (loading control) antibodies. A representative of five blots from five biological preparations is shown. Band intensities were quantified and relative amounts of $V_1$ were normalized against WT. Bottom panel, averages of relative band intensities from five experiments. Data are presented as mean ± SEM. Data information: Statistical significance (**A, C, D, E**) was calculated in GraphPad Prism 9 using unpaired Student's *t* test ($^{ns}$ indicates nonsignificant ($P > 0.05$); * indicates $P \leq 0.05$; ** indicates $P \leq 0.01$; *** indicates $P \leq 0.001$; **** indicates $P \leq 0.0001$). Source data are available online for this figure.

Oxr1p does not prefer inactive over active enzymes, contrary to our initial hypothesis. We therefore asked whether Oxr1p's ability to disassemble the V-ATPase in vitro plays a role in canonical V-ATPase regulation by reversible disassembly in vivo after all.

## Overexpression of *OXR1* lowers V-ATPase activity

To elucidate Oxr1p's physiological role in V-ATPase regulation, we deleted *OXR1* from wild-type yeast to generate strain *oxr1Δ*.

We then transformed *oxr1Δ* with a low-copy number plasmid (pRS316, CEN6, URA3) for expression of C-terminally HA-tagged Oxr1p (Oxr1p-HA) under its native promoter (strain *oxr1*$^{NP}$), or with a high-copy number plasmid (YEp352, 2 μ, URA3) for overexpression of Oxr1p-HA (strain *oxr1*$^{OE}$). Western blots of *oxr1*$^{NP}$ and *oxr1*$^{OE}$ lysates revealed a more than twofold overexpression of *OXR1* in the *oxr1*$^{OE}$ strain (Fig. 2A). Viability assays revealed that all three strains were able to grow at elevated pH in presence of $Ca^{2+}$, a condition that requires functional V-ATPase

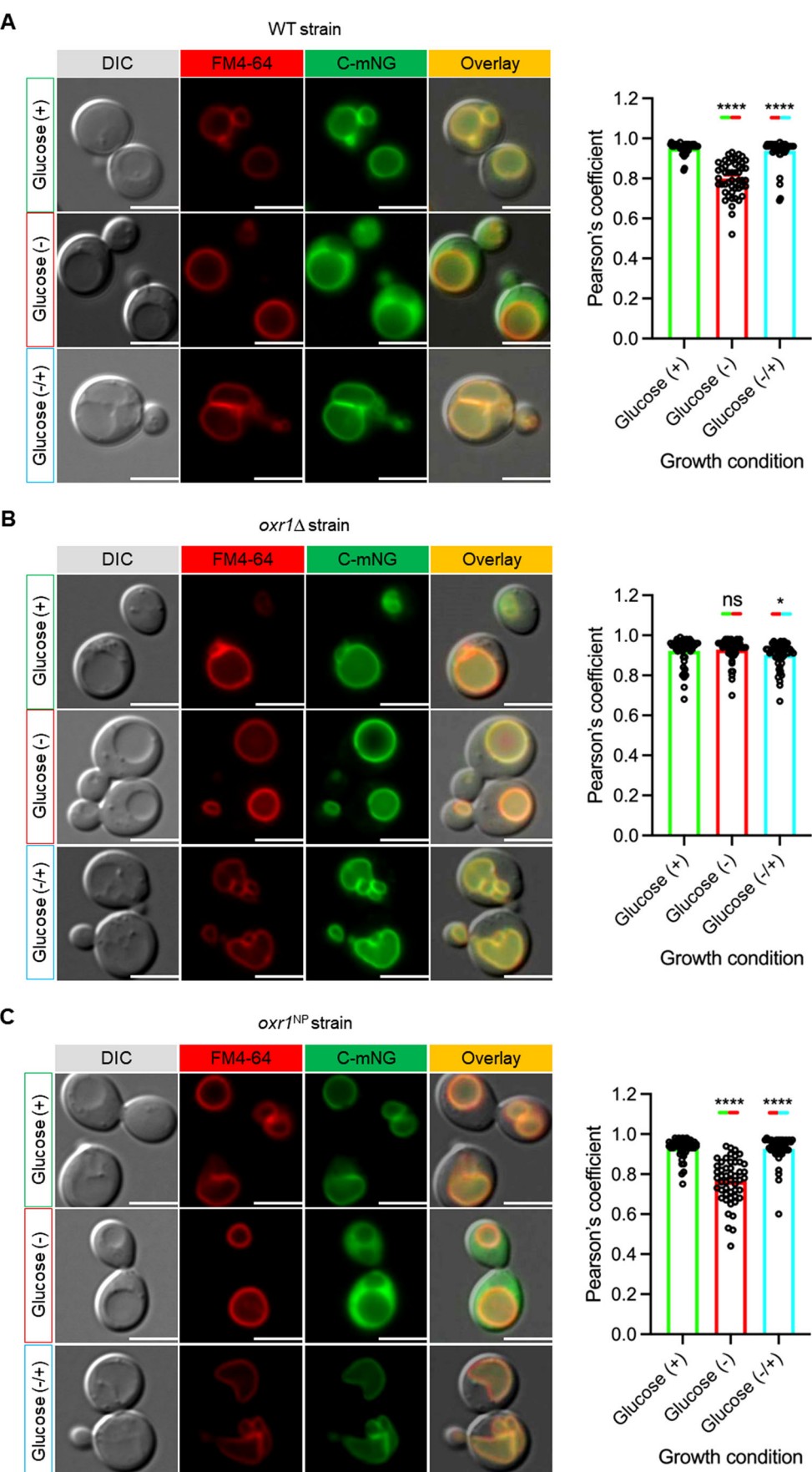

**Figure 3. The role of Oxr1p in reversible disassembly in vivo.**

(A–C) DIC and fluorescence images of cells expressing C-mNG (green) collected in glucose-fed (+), 15 min after glucose deprivation (−), and 15 min after glucose re-addition (−/+) conditions from (A) WT, (B) oxr1Δ, and (C) oxr1NP strains. The diffuse cytosolic C-mNG signal in the absence of glucose indicates disassembly of V-ATPase in WT and oxr1NP strains. However, in the oxr1Δ strain, C-mNG fluorescence co-localized with the vacuole-specific dye FM4-64 (red) when cells were starved of glucose. Bar graphs (right) show colocalization (Pearson's coefficient) of mNG-tagged C with FM4-64 under glucose-fed (green, +), deprived (red, −) and re-add (cyan, −/+) conditions. At least 50 cells (25 cells from each replicate) were used for the analysis in each condition. Data are plotted as mean ± SEM with individual points shown from two replicates. Scale bar: 5 μm. Data information: Statistical significance (A–C) was calculated in GraphPad Prism 9 using unpaired Student's $t$ test (ns indicates nonsignificant ($P > 0.05$); * indicates $P \le 0.05$; **** indicates $P \le 0.0001$). Source data are available online for this figure.

(Nelson and Nelson, 1990), with growth somewhat impaired for oxr1Δ and oxr1OE strains at elevated temperature (37 °C) (Fig. 2B). As a control, we used a yeast strain deleted for VMA1 (subunit A of $V_1$ subcomplex; vma1Δ), which displayed the typical V-ATPase loss of function, or Vma⁻ phenotype characterized by the inability to grow at elevated pH in presence of $Ca^{2+}$ (Fig. 2B). To verify that Oxr1p's activity is not impaired by the C-terminal HA tag, we bacterially expressed Oxr1p-HA and found it to be equally efficient in inhibiting the ATPase activity of purified vacuoles as the untagged protein (Appendix Fig. S2). Since the Vma⁻ phenotype is not obvious unless V-ATPase activity is reduced to less than ~30% of wild-type (Liu and Kane, 1996), we compared the ATPase activities of vacuoles isolated from the three mutant strains and wild-type. Whereas ATPase activities of oxr1Δ and wild-type vacuoles were similar, those from oxr1NP and oxr1OE strains were reduced to ~75 and ~50% of wild-type, respectively (Fig. 2C). To see whether this reduction in activity for the oxr1OE vacuoles was due to differences in V-ATPase assembly, we determined the relative ratios of $V_1$ over $V_o$ on purified vacuoles and found that whereas vacuoles from the oxr1Δ strain contained ~40% more $V_1$ compared to wild-type, oxr1OE vacuoles had ~25% less (Fig. 2D). However, western blot analysis of total cell lysates revealed comparable amounts of $V_1$ subunit A (Vma1p) in all strains, suggesting that $V_1$ expression is not affected by deletion or overexpression of OXR1 (Fig. 2E). Taken together, and considering that Oxr1p causes V-ATPase disassembly in vitro (Khan et al, 2022), the data suggest that the higher and lower levels of $V_1$ on oxr1Δ and oxr1OE vacuoles are due to *decreased* and *increased* V-ATPase *disassembly*, respectively.

## Oxr1p is required for efficient V-ATPase disassembly in vivo

V-ATPase reversible disassembly is a dynamic process, in which the balance between assembled and disassembled states is dictated by, for example, the glucose concentration (Parra and Kane, 1998). If Oxr1p is involved in the reversible disassembly process, this balance should be perturbed in its absence. To study the role of Oxr1p on V-ATPase disassembly in vivo, we generated a C-terminal fusion of $V_1$ subunit C with mNeonGreen (C-mNG) using homologous recombination in wild-type and oxr1Δ strains. The expectation here is that under conditions that favor assembly, most of the C-mNG will be part of holo V-ATPases, with fluorescence thus largely limited to the vacuole. On the other hand, in situations that favor disassembly, most of the C-mNG (and $V_1$) will be released into the cytoplasm, resulting in a diffuse fluorescence signal throughout the cell. To capture the reversible disassembly process, we collected fluorescence images from exponentially growing cells in three different growth conditions:

(i) glucose-fed, (ii) glucose starved, and (iii) glucose restored. In all experiments, vacuolar membranes were visualized using the vacuole-specific dye FM4-64, and colocalization of the C-mNG and FM4-64 fluorescence intensity was quantified via Pearson's correlation coefficients (Fig. 3). In wild-type cells, we observed the expected switch from C-mNG fluorescence on the vacuole under glucose-fed conditions, to diffuse fluorescence upon glucose withdrawal, and return of the signal to the vacuole upon glucose re-addition (Fig. 3A). Strikingly, the outcome of the experiment in the oxr1Δ cells was different: here the C-mNG fluorescence remained on the vacuole in all three conditions (Fig. 3B), indicating that Oxr1p is required for the enzyme disassembly step. To validate this result, we complemented the OXR1 deletion with a pRS316 plasmid expressing C-terminally HA-tagged Oxr1p and found that the switch of C-mNG fluorescence upon glucose withdrawal and re-addition was restored to wild-type levels (Fig. 3C). Combined with our earlier finding that Oxr1p causes V-ATPase disassembly in vitro (Khan et al, 2022), the above C-mNG fluorescence experiments provide strong evidence that Oxr1p is required for efficient V-ATPase disassembly in vivo. Next, we asked whether V-ATPase can disassemble in the oxr1Δ strain if we starve the cells of glucose longer than the 15 min used in our fluorescence microscopy assays. To answer this question, we collected fluorescence images of the oxr1Δ strain expressing C-mNG over a time period of 2 h at time points 0.5, 0.75, 1, 1.5, and 2 h following glucose deprivation (Fig. EV1). Whereas no significant disassembly was observed up to the 1 h time point, an increasing number of cells showed some level of cytosolic mNG fluorescence at 1.5 (23%) and 2 h (42%) (Fig. EV1B). The mechanism of this delayed (Oxr1p independent) disassembly in some of the cells, however, is currently not known and will require further investigation.

## Oxr1p is localized to the cytoplasm

A previous study had shown that C-terminally GFP-tagged Oxr1p localizes to mitochondria (Elliott and Volkert, 2004). However, algorithms for predicting mitochondrial targeting sequences (Fukasawa et al, 2015; Savojardo et al, 2020) did not detect such a sequence in Oxr1p's N-terminus, and available large-scale yeast protein localization studies using C-terminal GFP fusions were inconclusive (Huh et al, 2003). Moreover, the evidence presented above that Oxr1p is directly involved in reversible disassembly at the level of the enzyme means that Oxr1p must be able to access the vacuolar membrane under conditions that promote V-ATPase disassembly. To revisit the cellular localization of Oxr1p, we generated a genomic C-terminal fusion of Oxr1p with mNeon-Green (Oxr1p-mNG) using homologous recombination. Of note, our recent structure of $V_1$(C)Oxr1p shows Oxr1p's C-terminus partially buried in the $V_1$:Oxr1p interface (Khan et al, 2022), which

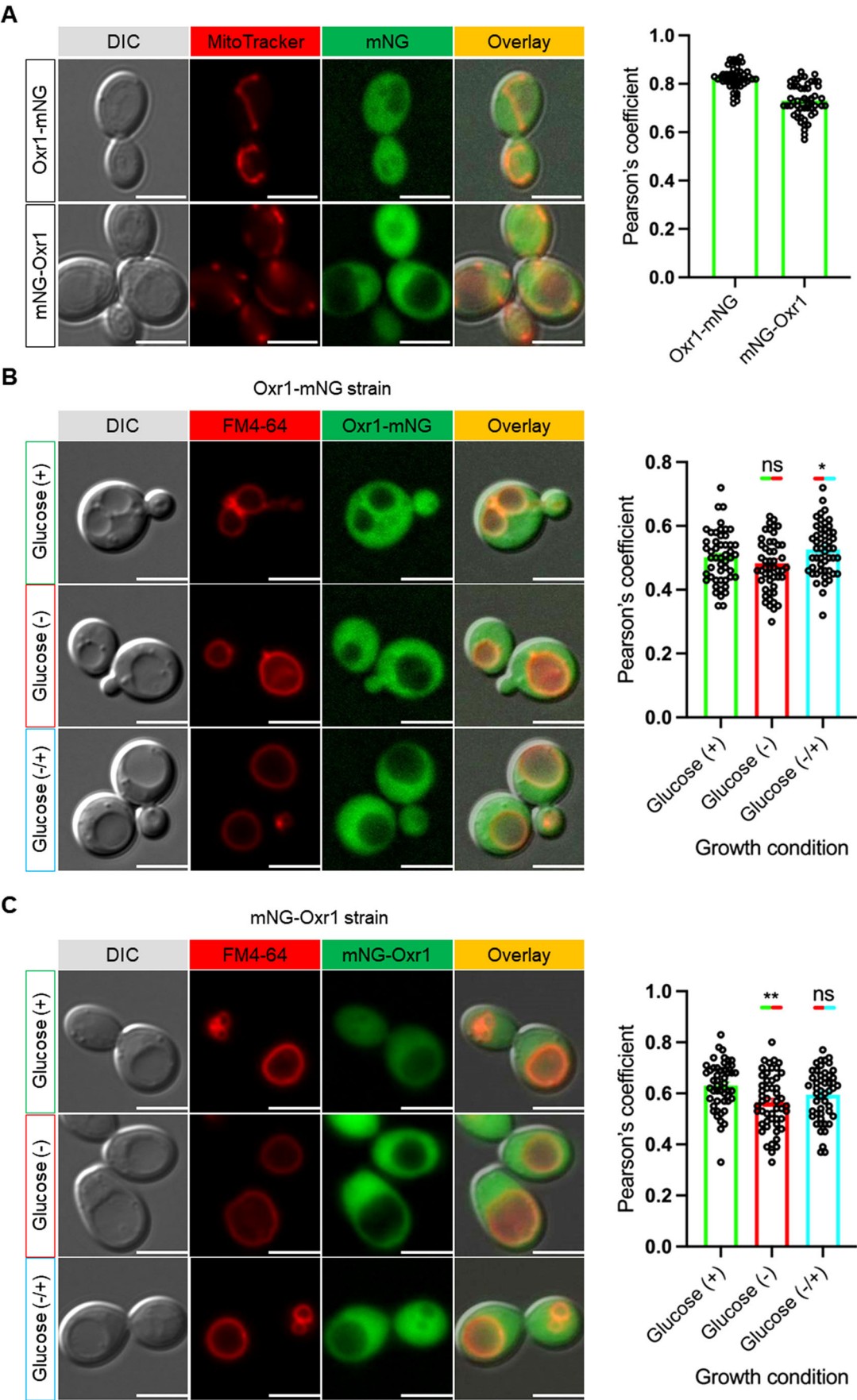

**Figure 4.  Oxr1p is localized to the cytosol.**

(A) DIC and fluorescence images of exponentially growing yeast cells expressing Oxr1p fused with mNG either at the C-terminus (Oxr1p-mNG) or N-terminus (mNG-Oxr1p). As a control, MitoTracker red dye was used to visualize mitochondria. Around ~100 cells (50 cells from each replicate) were analyzed. Bar graph (right) shows colocalization (Pearson's coefficient) of mNG-tagged Oxr1p with MitoTracker dye. At least 50 cells (25 cells from each replicate) were used for the analysis in each strain. Data are plotted as mean ± SEM with individual points shown from two replicates. Scale bar: 5 μm. (B, C) DIC and fluorescence images of yeast cells expressing C- (B) or N-terminally (C) mNG-tagged Oxr1p recorded under glucose-fed (+), 15 min after glucose deprivation (−) and 15 min after glucose re-addition (−/+) conditions (green). Vacuoles are stained with FM4-64 (red). Bar graphs (right) show colocalization (Pearson's coefficient) of mNG-tagged Oxr1p with FM4-64 under glucose-fed (green, +), deprived (red, −) and re-add (cyan, −/+) conditions. At least 50 cells (25 cells from each replicate) were used for the analysis in each condition. Data are plotted as mean ± SEM with individual points shown from two replicates. Scale bar: 5 μm. Data information: Statistical significance (B, C) was calculated in GraphPad Prism 9 using unpaired Student's *t* test (ns indicates nonsignificant ($P > 0.05$); * indicates $P \leq 0.05$; ** indicates $P \leq 0.01$). Source data are available online for this figure.

means that a bulky C-terminal fusion such as a fluorescent protein (~27 kDa) could impair Oxr1p's ability to bind the enzyme. We therefore transformed the *oxr1Δ* background with a plasmid expressing N-terminally mNG-tagged Oxr1p (mNG-Oxr1p). To visualize mitochondria, we used MitoTracker dye. The fluorescence microscopy images show diffuse mNG fluorescence throughout the cytoplasm for both strains, with some mNG fluorescence co-localizing to MitoTracker dye fluorescence as quantified using the Pearson's coefficient (Fig. 4A). Thus, whereas the experiment suggests that some Oxr1p may be targeted to mitochondria, the images show that the bulk of mNG-tagged Oxr1p is cytosolic, consistent with Oxr1p's ability to bind and disassemble the V-ATPase on the vacuolar membrane in vivo. Given its cytoplasmic localization, we then asked whether Oxr1p is recruited to the vacuolar membrane upon glucose withdrawal, analogous to the V-ATPase assembly chaperone, RAVE, which goes to the vacuole upon glucose re-addition (Jaskolka and Kane, 2020). To address this question, we collected fluorescence images of both strains expressing mNG-tagged Oxr1p in the three different growth conditions as described in the previous section: (i) glucose-fed, (ii) glucose starved, and (iii) glucose re-addition. However, unlike what was observed for RAVE, both constructs remained equally distributed in between vacuole and cytoplasm in all three conditions, with only minor changes for Oxr1p-mNG and mNG-Oxr1p localization upon glucose re-addition and withdrawal, respectively (Fig. 4B,C).

## Efficient V-ATPase disassembly by Oxr1p is ATP-dependent

Prior research using live-cell imaging showed that glucose withdrawal-induced disassembly of yeast V-ATPase occurs on a timescale of ~5–10 min (Dechant et al, 2010), consistent with the live-cell microscopy data for wild-type and *oxr1*[NP] strains (Fig. 3A,C). This is in stark contrast to our recent study, where we found that ~30% disassembly of in vitro reconstituted V-ATPase required a 16-h incubation with a twofold molar excess of Oxr1p (Khan et al, 2022). In vitro reconstitution of V-ATPase from individual subcomplexes ($V_1$, $V_o$, and C), however, requires replacing of the inhibitory H subunit with a *chimeric* version of H ($H_{chim}$) composed of yeast N-terminal ($H_{NT}$) and human C-terminal ($H_{CT}$) domains (Sharma et al, 2019). While the $H_{chim}$ containing V-ATPase is active and fully coupled, it resists the slow ATP hydrolysis-induced disassembly observed for the wild-type enzyme (Sharma et al, 2019; Sharma and Wilkens, 2017). We therefore wondered whether the slower disassembly rate observed for the in vitro assembled enzyme was due to the presence of $H_{chim}$.

To address this question, we affinity-purified wild-type V-ATPase and reconstituted the complex into lipid nanodiscs (Appendix Fig. S3). The purified enzyme was then incubated with a twofold molar excess of Oxr1p and measured the time-dependent inhibition of ATPase activity over 16 h. However, similar to the $H_{chim}$ containing enzyme, wild-type V-ATPases retained around 70% and 30% of ATPase activity after 30 min and 16 h of incubations, respectively (Fig. 5A). Thus, whereas the live-cell fluorescence microscopy data presented in the previous sections provided strong evidence that Oxr1p plays a key role in reversible disassembly, the rate of the process observed in vivo vs in vitro is significantly faster, suggesting the involvement of additional factor(s) in the cell.

As discussed earlier, efficient V-ATPase disassembly in vivo requires enzymatic activity (ATP hydrolysis) (Liu and Kane, 1996; Parra and Kane, 1998). However, which step of the disassembly process is ATP-dependent is not known. We recently reported that the dissociation of mutant $V_1$ subcomplexes from immobilized subunit C is greatly accelerated in the presence of ATP (Sharma et al, 2019). We therefore asked whether ATP might affect the rate of Oxr1p-mediated disassembly. To address this question, we incubated wild-type V-ATPase with a twofold molar excess of Oxr1p or ATP individually or in combination for 30 min, and measured ATPase activities. Whereas Oxr1p alone resulted in ~30% loss of activity, inhibition was close to complete in presence of Oxr1p and ATP (Fig. 5B), with negative stain EM confirming virtually complete V-ATPase disassembly (Fig. 5C,D). Thus, while Oxr1p alone is able to slowly disassemble V-ATPase, the presence of ATP makes the process much faster. To better understand the kinetics of Oxr1p-mediated V-ATPase disassembly, we carried out biolayer interferometry (BLI) experiments using purified V-ATPase immobilized on streptavidin biosensors via biotinylated membrane scaffold protein (bio-MSP) that was used to reconstitute V-ATPase. V-ATPase-loaded sensors were exposed to Oxr1p, or nucleotides, or combinations thereof, and the release of $V_1$ was monitored via the time-dependent decrease in BLI signal. Consistent with ATPase activity and negative stain EM analysis (Fig. 5A–D), Oxr1p or ATP alone caused only partial release of $V_1$ over the duration of the experiment, with their combination reaching almost 70% disassembly within 5 min (Fig. 5E), a rate similar to the one observed in vivo (Dechant et al, 2010). In contrast, the non-hydrolysable ATP analog, AMP-PNP, did not accelerate $V_1$ release in the presence of Oxr1p, consistent with the previously stated observation that ATP hydrolysis is required for efficient V-ATPase disassembly in the cell. Overall, these results provide strong evidence that whereas either Oxr1p or ATP alone promotes V-ATPase disassembly, the process is much faster and more efficient when Oxr1p and ATP work together.

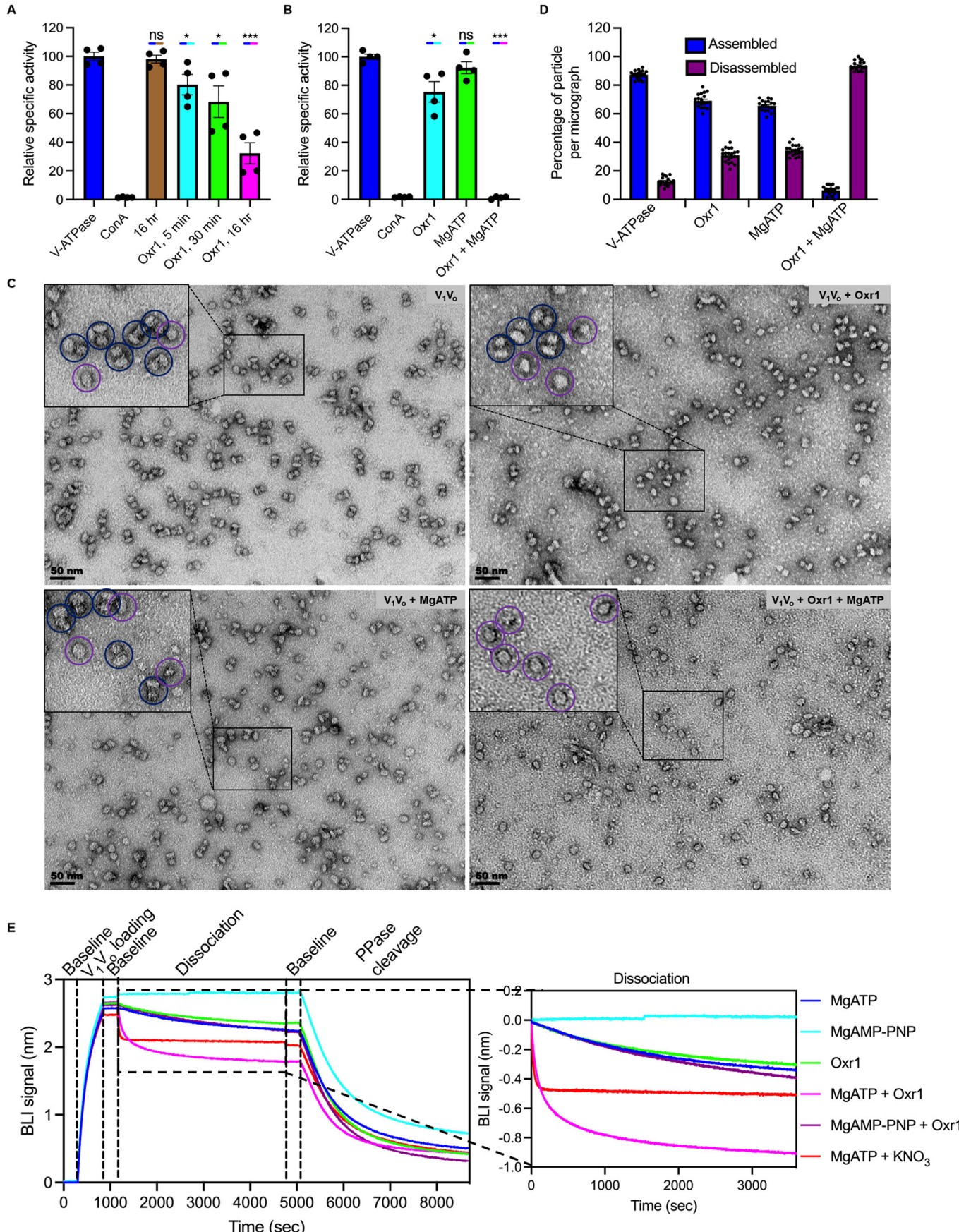

**Figure 5. Efficiency of Oxr1p-mediated V₁ disassembly from purified lipid nanodisc reconstituted V-ATPase in the presence of nucleotides.**

(A) ATPase activities of purified V-ATPase pre-incubated with a twofold molar excess of Oxr1p for the indicated times. V-ATPase incubated for 16 h in the absence of Oxr1p served as control (brown). Relative activities were normalized against the starting V-ATPase activity (blue). ConA served as an additional control. Individual data points of four tests from two biological preparations are shown. Data are presented as mean ± SEM. (B) ATPase activities of purified V-ATPase pre-incubated for 30 min with a twofold molar excess of Oxr1p in the absence (cyan) or presence (magenta) of 4 mM MgATP. As a control, ATPase activity was measured in the presence of 4 mM MgATP only (green). Relative activities were normalized against the starting V-ATPase activity (blue). Activity in the presence of ConA served as an additional control. Individual data points of four tests from two biological preparations are shown. Data are presented as mean ± SEM. (C) Negative stain electron micrographs of samples from (B). Insets, assembled (V₁Vₒ) and disassembled (V₁ or Vₒ) particles in each condition were identified by their shape as highlighted by blue and purple circles, respectively. (D) The numbers of particles representing assembled (blue) and disassembled (purple) complexes are given as percentages of total particles for each of 20 micrographs. Individual data points are plotted and presented as mean ± SEM. A representative analysis of two tests from two biological preparations is shown. (E) Kinetics of Oxr1p-mediated disassembly of V₁ in the presence of nucleotides probed by BLI. V-ATPase was immobilized on streptavidin biosensors via biotinylated nanodiscs and kinetics of V₁ dissociation from Vₒ was monitored in either (1) 1 mM MgATP (blue), (2) 1 mM MgAMP-PNP (cyan), (3) 1 μM Oxr1p (green), (4) 1 mM MgATP + 1 μM Oxr1p (magenta), (5) 1 mM MgAMP-PNP + 1 μM Oxr1p (purple), and (6) 1 mM MgATP + 300 mM KNO₃ (red). The inset shows the enlarged view of the V₁ dissociation step. A representative of three tests from two biological preparations is shown. Data information: Statistical significance (A, B) was calculated in GraphPad Prism 9 using unpaired Student's t test (ⁿˢ indicates nonsignificant (P > 0.05); * indicates P ≤ 0.05; *** indicates P ≤ 0.001). Source data are available online for this figure.

While we find here that efficient Oxr1p-mediated disassembly of V₁ from purified wild-type V-ATPase requires the presence of ATP, the process appeared much faster with purified vacuoles even in the absence of added nucleotides (Khan et al, 2022). One explanation for this discrepancy is that the molar ratio of Oxr1p to V-ATPase was likely much higher in these experiments, as the ratio was based on total vacuolar protein, of which V-ATPase is only 10–25% based on specific ConA sensitive ATPase activities of vacuoles ($\sim$1–2.5 μmol × (min × mg)$^{-1}$) compared to purified enzyme ($\sim$10 μmol × (min × mg)$^{-1}$) (Appendix Fig. S3). To validate this explanation, and to test whether nucleotide could make the process even more efficient with vacuoles as we observed with purified enzyme, we incubated vacuoles with Oxr1p at ratios of 1:10 or 0.3:10 (w/w) over total vacuolar protein in presence and absence of ATP for 30 min and measured ATPase activities. While incubation with ATP alone caused a slight increase of the activity, a ratio of 1:10 (w/w) Oxr1p over vacuolar protein resulted in an almost complete loss of activity (but not vacuolar V₁, see below) as observed in our previous study (Khan et al, 2022) (Fig. EV2A). However, when Oxr1p was added at lower concentration (0.3:10 (w/w) over vacuolar protein), vacuoles retained close to 50% of the activity (Fig. EV2A). Strikingly, when combining 0.3:10 (w/w) Oxr1p with ATP, we saw an almost complete loss of activity after 30-min incubation (Fig. EV2A), indicating that Oxr1p is equally efficient at disassembling the V-ATPase on vacuoles than for the purified enzyme in lipid nanodiscs.

We previously found that whereas treatment of vacuoles with Oxr1p resulted in almost complete loss of ATPase activity, release of V₁ was incomplete (Khan et al, 2022). Here, using western blot analysis, we expanded on these earlier experiments and found that neither increasing the amount of added Oxr1p nor the incubation time resulted in more V₁ release (Fig. EV2B,C), suggesting that a subset of the V-ATPases were resistant to Oxr1p-induced disassembly. To test whether nucleotides could make enzyme disassembly from vacuoles more efficient, as seen for the purified enzyme (Fig. 5D,E), we incubated vacuoles with a 0.3:10 (w/w) ratio of Oxr1p over vacuolar protein alone or in the presence of ATP or AMP-PNP. The western analysis showed that whereas Oxr1p alone caused only partial release of V₁ subcomplexes from the membrane, in the presence of ATP, but not AMP-PNP, almost all of the V₁ was found in the supernatant fraction (Fig. EV2D). To summarize, ATP enhances Oxr1p's ability to disassemble the enzyme on vacuoles, mirroring the effect of ATP on Oxr1p's ability to disassemble the

purified enzyme as measured using negative stain EM (Fig. 5C,D) and BLI (Fig. 5E).

## ATP hydrolysis breaks the Oxr1p-V₁ interaction

Given Oxr1p causes release of wild-type V₁ from Vₒ, and that wild-type V₁ does not co-purify with Oxr1p (Kitagawa et al, 2008) we wondered, what happens to Oxr1p following V-ATPase disassembly? The endogenous Oxr1p-bound mutant V₁ subcomplex (V₁(C) Oxr1p) resolved by cryoEM was derived from a yeast strain in which V₁ subunit H was deleted (vma13Δ) (Khan et al, 2022) (Figs. 1B and 6A). To determine the fate of Oxr1p in yeast strains expressing wild-type V-ATPase, we performed immunoprecipitation (IP) experiments by pulling down Oxr1p or subunit A (Vma1p) from oxr1ᴼᴱ and oxr1Δ strain lysates and testing for co-precipitation of subunit A and Oxr1p, respectively. Surprisingly, V₁ and Oxr1p did not Co-IP from the oxr1ᴼᴱ lysate (Fig. 6B), suggesting that either Oxr1p does not remain bound to wild-type V₁ after enzyme dissociation, or that the V₁-Oxr1p interaction is broken upon preparation of the cytosolic lysate. To resolve this discrepancy, we tested whether recombinant biotinylated Oxr1p (bio-Oxr1p) can pull down purified wild-type V₁ or V₁ that is contained in cytosolic lysate from the oxr1Δ strain. However, in both cases, we were not able to detect an interaction between Oxr1p and wild-type V₁ (Fig. EV3A,B). Prior to the above experiments, we tested bio-Oxr1p's ability to inhibit V-ATPase on purified vacuoles and found it to be equally effective as the non-biotinylated protein (Appendix Fig. S2). In summary, whereas Oxr1p inhibits the MgATPase activity of mutant V₁ subcomplexes and forms a stable complex with V₁ΔH and C as seen by cryoEM (Khan et al, 2022), above experiments show that Oxr1p does not bind wild-type V₁ under the conditions tested. Next, since the V₁(C)Oxr1p complex solved by cryoEM was isolated from a subunit H deletion strain (vma13Δ) (Khan et al, 2022), we determined whether we could Co-IP Oxr1p and V₁ΔH from vma13Δ lysate. To do so, we deleted OXR1 from the vma13Δ strain (resulting in oxr1Δvma13Δ) and then complemented oxr1Δ with Oxr1p-HA via the pRS316 plasmid to generate strain oxr1ᴺᴾvma13Δ. Subsequent experiments using cleared lysate of the oxr1ᴺᴾvma13Δ strain showed that whereas a pulldown of V₁ΔH co-precipitated Oxr1p-HA (Fig. 6C), IP of Oxr1p via its C-terminal HA tag failed to pull down V₁ΔH. As mentioned above, Oxr1p's C-terminus is partially buried in the V₁:Oxr1p interface, and it is possible that the α-HA antibody

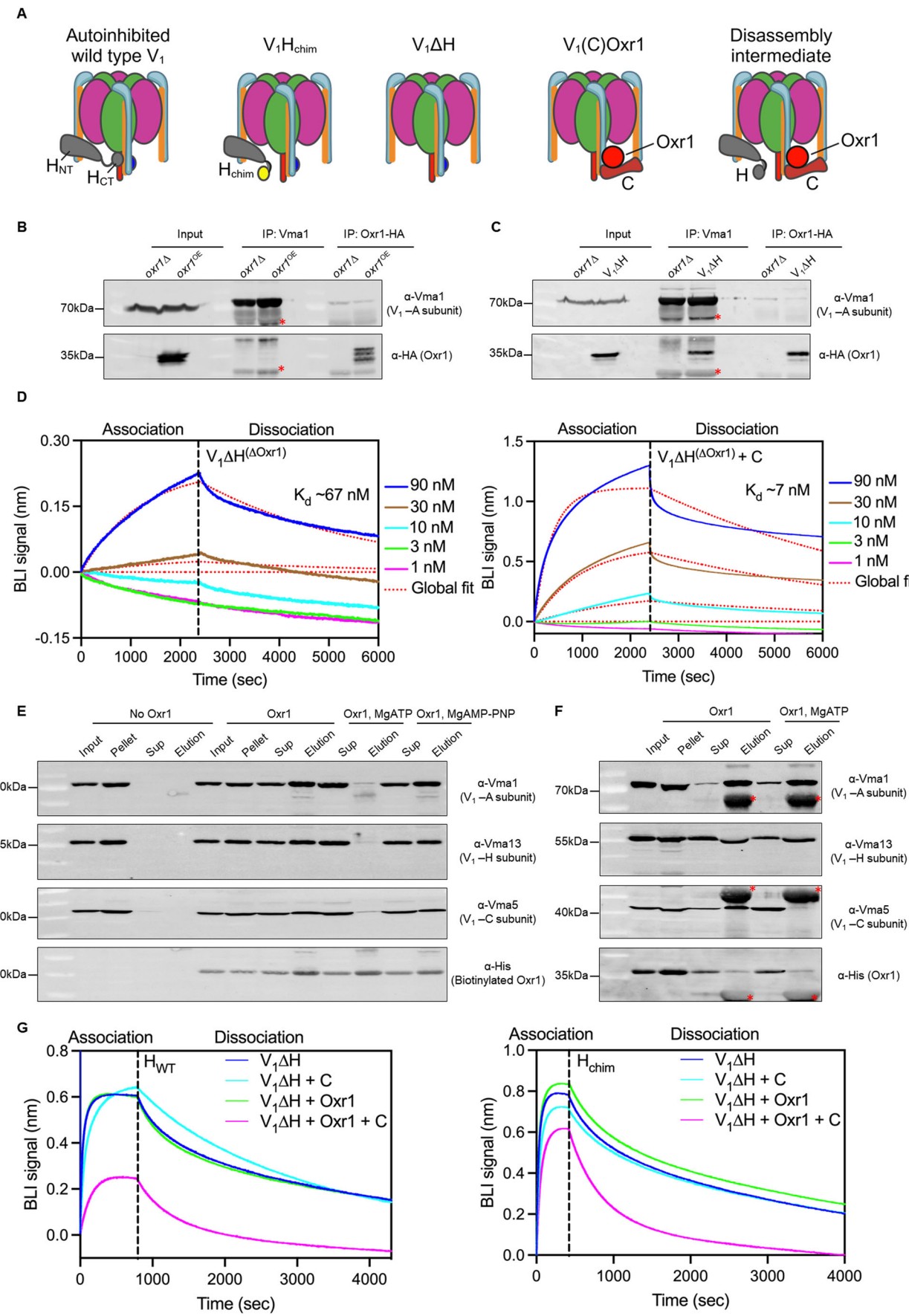

◀  **Figure 6.  MgATP hydrolysis by the "disassembly intermediate" V₁(C,H)Oxr1p releases Oxr1p and subunit C.**

(A) Cartoon representation of the different $V_1$ subcomplexes considered here. Wild-type $V_1$ has H bound in its autoinhibited conformation with $H_{NT}$ bound to EG1 and $H_{CT}$ interacting with EG2. $V_1H_{chim}$ has MgATPase activity as $H_{chim}$ is not in the autoinhibitory conformation. $V_1\Delta H$ has MgATPase activity. The $V_1(C)Oxr1p$ complex lacks subunit H as it was obtained from the *vma13Δ* strain. Subunit H in the $V_1(C,H)Oxr1p$ disassembly intermediate is not in its autoinhibited conformation. The figure was created with BioRender.com. (B) Western blot of α-Vma1p (pulling down wild-type $V_1$) and α-HA (pulling down Oxr1p) IP using cleared lysates of the *OXR1* overexpressing strain (*oxr1^{OE}*) and the *oxr1Δ* strain as a control. A representative blot from three experiments is shown. (C) Western blot of α-Vma1p (pulling down $V_1\Delta H$) and α-HA (pulling down Oxr1p) IP from cleared lysates of the strain deleted for subunit H and expressing Oxr1p-HA from a plasmid under the native promoter (*oxr1^{NP}vma13Δ*) and the *oxr1Δ* strain as a control. A representative blot from three experiments is shown. (D) BLI analysis of the interaction between immobilized bio-Oxr1p and $V_1\Delta H$ purified from the $V_1\Delta H^{(\Delta Oxr1)}$ strain in the absence (left) and presence (right) of subunit C. Global fitting of the BLI traces obtained in the presence of C gave a $K_d$ of ~7 ± 1 nM (right panel). Note that in the absence of C (left panel), global fitting was less reliable due to the relatively low BLI signal combined with significant non-specific binding of $V_1\Delta H$ to empty sensors, with a resulting estimate of the $K_d$ of ~67 ± 33 nM. A representative of three experiments (only association and dissociation steps) from two biological preparations is shown. (E) Western blot of a pulldown of disassembled $V_1$ complexes in the absence or presence of the indicated nucleotides using bio-Oxr1p as bait. Purified vacuoles were incubated with or without bio-Oxr1p for 30 min before ultracentrifugation. Supernatants were then incubated either with buffer or nucleotides for an additional 30 min before bio-Oxr1p pull down via streptavidin beads for 1 h (see "Methods" for details). The antibodies used for detection were α-Vma1p ($V_1$ subunit A), α-Vma13p ($V_1$ subunit H), α-Vma5p ($V_1$ subunit C), and α-5xHis (bio-Oxr1p). The immunoblot analysis suggests that Oxr1p remains bound to $V_1$ in the absence of ATP hydrolysis as evidenced from the presence of $V_1$ subunits A, H and C along with Oxr1p in the *elution* fraction in both Oxr1p and Oxr1p + AMP-PNP conditions. However, when ATP is present, the interaction between Oxr1p and $V_1$ gets broken as the *elution* fraction in the Oxr1p + ATP condition does not have any of the bands for the $V_1$ subunits, but Oxr1p. Note that due to the incomplete biotinylation of Avi-tagged Oxr1p (bio-Oxr1p), the sup fractions of bio-Oxr1p treated conditions have bands for bio-Oxr1p and $V_1$ subunits that are not pulled down by streptavidin beads. A representative of three experiments from two biological preparations is shown. (F) Western blot of a reverse pulldown of disassembled $V_1$ complexes in the absence or presence of ATP using FLAG-Vma10p (N-terminally FLAG-tagged $V_1$ subunit G) as bait. The antibodies used for detection were as in (E). The immunoblot suggests that along with Oxr1p, subunit C also gets released from the disassembled $V_1$ following ATP hydrolysis. In addition, while subunit H remains bound to $V_1$ after ATP hydrolysis, Oxr1p can lead to release of some H in the absence of ATP as seen in the *sup* fraction. A representative of three experiments from two biological preparations is shown. (G) BLI analysis of the interactions between immobilized subunits H (left panel) and $H_{chim}$ (right panel) with $V_1\Delta H$ monitored in the presence and absence of Oxr1p and subunit C. The BLI signal shows that in the presence of subunit C and Oxr1p, binding of $V_1\Delta H$ to wild type H is reduced by ~60% (magenta) compared to the Oxr1p only condition (green) (left panel), with the inhibition much less prominent in case of $H_{chim}$ (~25%) (right panel). A representative of three experiments (only association and dissociation steps) from two biological preparations is shown. Data information: Asterisk (*) (B, C, F) indicates the heavy and/or light chain of antibodies used in the pulldown. Source data are available online for this figure.

therefore only binds the excess of Oxr1p-HA that is not attached to $V_1\Delta H$. To verify this suspicion, we performed pulldown and BLI experiments with $V_1\Delta H$ and N-terminally tagged Oxr1p (bio-Oxr1p). To get an accurate measure of $V_1\Delta H$'s affinity for Oxr1p, we purified $V_1\Delta H$ from the *oxr1Δvma13Δ* double-deletion strain (hereafter referred to as $V_1\Delta H^{(\Delta Oxr1)}$) to avoid contamination with endogenous Oxr1p. $V_1\Delta H^{(\Delta Oxr1)}$ is active and has a SEC elution profile similar to $V_1\Delta H$ purified from the *vma13Δ* strain (Fig. EV3C). Consistent with our explanation that IP of Oxr1p-HA does not co-precipitate $V_1\Delta H$ due to steric hindrance, we find that bio-Oxr1p is able to pull down purified $V_1\Delta H^{(\Delta Oxr1)}$ as analyzed by Coomassie blue stained SDS-PAGE (Fig. EV3D). The interaction between Oxr1p and $V_1\Delta H^{(\Delta Oxr1)}$ was further analyzed using biolayer interferometry (BLI). Here, bio-Oxr1p was immobilized on streptavidin sensors and the sensors were then dipped into solutions of $V_1\Delta H^{(\Delta Oxr1)}$ in the absence and presence of subunit C (Fig. 6D) as subunit C was previously shown to enhance Oxr1p's ability to inhibit MgATPase activity of mutant $V_1$ subcomplexes (Khan et al, 2022). From the BLI experiments, we find that bio-Oxr1p binds $V_1\Delta H^{(\Delta Oxr1)}$ both in absence and presence of subunit C, with $K_{ds}$ of ~67 ± 33 nM and ~7 ± 1 nM, respectively (Fig. 6D), consistent with our earlier ATPase inhibition assays and the stability of the $V_1(C)Oxr1p$ complex (Khan et al, 2022).

The above experiments indicated that Oxr1p's ability to bind $V_1$ is controlled by the presence of subunit H, which is known to be required for autoinhibition of membrane detached wild-type $V_1$ (Parra et al, 2000). Previous structural studies had shown that enzyme disassembly involves a conformational switch of subunit H's C-terminal domain ($H_{CT}$) from a binding site on the N-terminal domain of $V_o$ subunit *a* in holo V-ATPase to its autoinhibitory binding site on $V_1$ (Oot et al, 2016; Vasanthakumar et al, 2022). Since we find that Oxr1p binds $V_1\Delta H$, but not wild-type autoinhibited $V_1$, we hypothesized that following

Oxr1p-induced V-ATPase disassembly, Oxr1p remains bound to $V_1$ until the complex adopts its autoinhibited conformation. To test this prediction, we incubated purified wild-type vacuoles with bio-Oxr1p to induce V-ATPase disassembly and then isolated the resulting $V_1$ subcomplex for pulldown (using bio-Oxr1p as bait) and western blot analysis. In support of our hypothesis, we found that following Oxr1p-induced V-ATPase disassembly, Oxr1p remains bound to $V_1$. Unexpectedly, the experiment showed that both subunits C and H were also part of the disassembled $V_1$ subcomplex next to Oxr1p (Fig. 6E), an observation that appeared to contradict our earlier observation that Oxr1p does not bind autoinhibited wild-type (H containing) $V_1$. The presence of subunit C in the membrane detached $V_1$, however, indicated that this complex must represent a *disassembly intermediate*, $V_1(C,H)Oxr1p$, as subunit C is known to dissociate from $V_1$ during enzyme disassembly in vivo (Liu and Kane, 1996). To resolve this seeming contradiction, we surmised that $V_1(C,H)Oxr1p$ is not (yet) autoinhibited, and therefore still able to hydrolyze ATP. We previously reported that the interaction between immobilized C subunit and an active mutant $V_1$ subcomplex ($V_1H_{chim}$) is disrupted upon ATP hydrolysis (Sharma et al, 2019), and that Oxr1p's ability to inhibit mutant $V_1$ subcomplexes ($V_1\Delta H$ and $V_1H_{chim}$) is C dependent (Khan et al, 2022). This means that if $V_1(C,H)Oxr1p$ is capable of hydrolyzing ATP, the accompanying conformational changes could then facilitate release of subunit C and Oxr1p and allow $H_{CT}$ to assume its autoinhibitory conformation. To test this prediction, we again incubated wild-type vacuoles with bio-Oxr1p and treated the resulting $V_1(C,H)Oxr1p$ disassembly intermediate with ATP or AMP-PNP before pulldown. The western blot revealed that ATP, but not AMP-PMP, disrupted the interaction between Oxr1p and $V_1$ (Fig. 6E). Whether subunits H and/or C were also released under these conditions, however, was not clear. To answer this question, we carried out reverse pulldowns

                                                               

with $V_1$ as bait and found that in the presence of ATP, both Oxr1p and C were released from the disassembly intermediate, but not H (Fig. 6F), consistent with H being required for $V_1$ autoinhibition. In the recent cryoEM structure of autoinhibited $V_1$, $H_{CT}$ is seen in contact with the N-termini of EG2 (Vasanthakumar et al, 2022), a conformation that requires less *bending* of EG2 than seen in the structure of the $V_1$(C)Oxr1p complex (Khan et al, 2022) (Appendix Fig. 4A). Since the autoinhibitory conformation of subunit H is mediated by $H_{CT}$, this suggests that presence of Oxr1p and C interferes with $H_{CT}$ binding to $V_1$. Thus, EG2 can either support binding of Oxr1p + C in $V_1$(C)Oxr1p (and $V_1$(C,H)Oxr1p), or $H_{CT}$ in autoinhibited $V_1$, but not both, which explains why Oxr1p does not bind autoinhibited $V_1$. To further support our model, we tested the interaction of immobilized subunit H with $V_1\Delta H$ in the absence and presence of Oxr1p and subunit C using BLI. The data shows that binding of $V_1\Delta H$ to wild-type H is significantly impaired by the presence of Oxr1p and C (Fig. 6G, left panel). However, when the experiment was carried out using $H_{chim}$, the interference was minimal (Fig. 6G, right panel). $H_{chim}$, which is not autoinhibitory but supports in vitro assembly of functional V-ATPase (Oot et al, 2016; Sharma et al, 2019), has been proposed to bind $V_1$ via $H_{NT}$ only, unlike wild-type H that binds via both $H_{NT}$ and $H_{CT}$ (Oot et al, 2016; Sharma et al, 2018; Vasanthakumar et al, 2022) (Fig. 6A). Overall, these results support a model in which Oxr1p-mediated V-ATPase disassembly first produces a $V_1$ subcomplex that has both Oxr1p and subunit C bound, and that subsequent ATP hydrolysis by this disassembly intermediate allows $H_{CT}$ to adopt its autoinhibitory

conformation, with concomitant release of Oxr1p and subunit C (see also Movie EV1).

## Oxr1p binds to holo V-ATPase in rotary state 1

Pioneering cryoEM structural work by the Rubinstein lab showed that purified yeast V-ATPase vitrified in the absence of substrate can be sorted into three major classes distinguished by three angular positions of the central rotor separated by 120° (Zhao et al, 2015). The three classes are referred to as rotary states 1–3, with state 1 being the most and state 3 being the least populated. Our recent cryoEM structure of $V_1\Delta H$ bound to Oxr1p and subunit C ($V_1$(C)Oxr1p) showed the complex to be halted in rotary state 1, with Oxr1p interacting with the C-terminal domain of a B subunit of the open catalytic site (Khan et al, 2022). How Oxr1p binds holo V-ATPase, however, is not known. To address this question, we incubated lipid nanodisc reconstituted yeast V-ATPase with a twofold molar excess of Oxr1p and imaged the resulting mixture using cryoEM. We used 3-D classification to obtain two major classes of particles, with the final maps determined at ~8.2 Å resolution. Strikingly, one of the maps showed a prominent density above the C subunit next to EG2 that resembled Oxr1p (Fig. 7A, see the pink dashed circle in the bottom left panel). The second class on the other hand had no significant density at that position (Fig. 7B, see pink dashed circle in bottom right panel). Comparison of the two maps to our recent models of the yeast enzyme in states 1–3 (Khan et al, 2022) showed that Oxr1p-bound V-ATPase is in rotary state 1 (~31% of the dataset) (Fig. 7A), with the second class

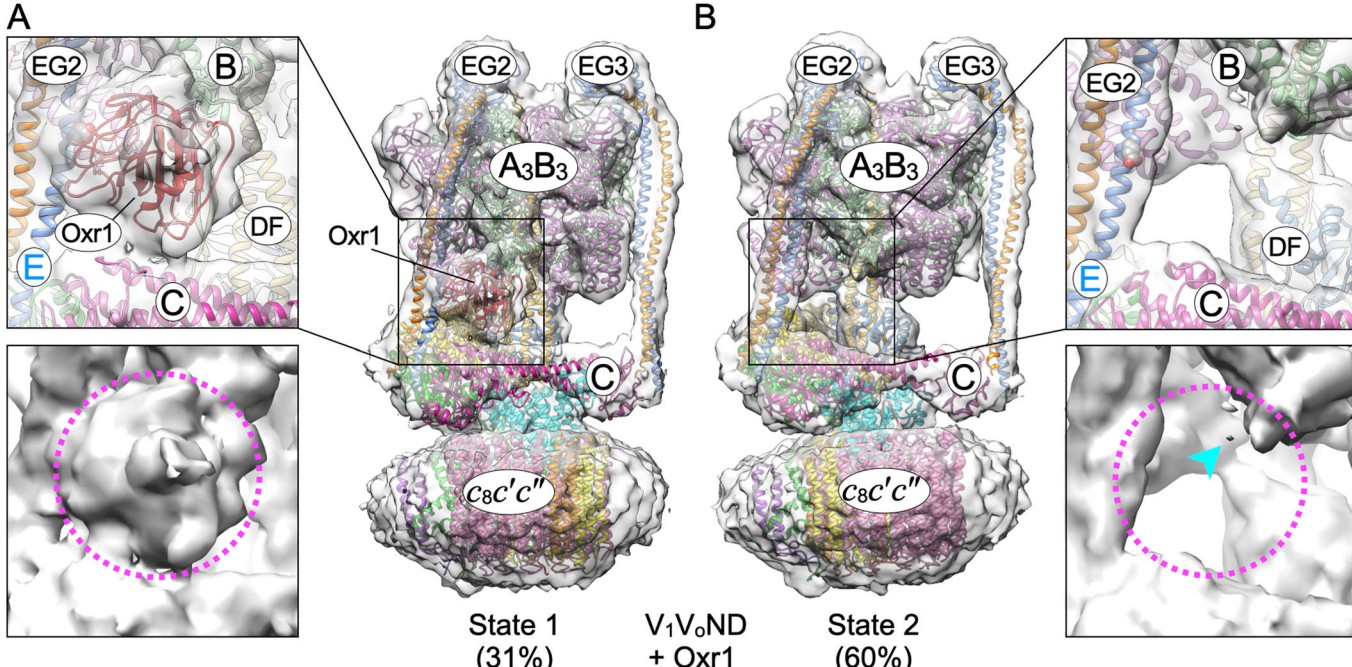

**Figure 7. CryoEM structures of lipid nanodisc reconstituted V-ATPase ($V_1V_0$ND) with bound recombinant Oxr1p.**

(A, B) CryoEM of yeast V-ATPase vitrified in presence of recombinant Oxr1p. 3-D classification of a dataset of 23,251 particle images revealed two major classes, with corresponding maps resolved at ~8.2 Å. (A) The class 1 map contains ~31% of the particles and shows the V-ATPase in rotary state 1, with extra density matching Oxr1p near EG2, B, and the C subunit. (B) The class 2 map contains 60% of the particle images and shows the enzyme in rotary state 2. The weak density near EG2, B, and C (see cyan arrowhead) could be due to imperfection in 3-D classification, or some weak, transient interaction between Oxr1p and the state 2 enzyme.

matching rotary state 2 (~60%) (Fig. 7B) (the remaining ~9% of the particles did not allow unambiguous assignment to either class). This suggests that Oxr1p-induced V-ATPase disassembly starts with the enzyme in state 1, and that subsequent ATP hydrolysis (at least one 120° rotary step) allows the resulting $V_1$(C,H)Oxr1p disassembly intermediate to adopt the autoinhibited state 2 conformation with concomitant release of Oxr1p and C (see "ATP hydrolysis breaks the Oxr1p-$V_1$ interaction" above). We also obtained a dataset of V-ATPases vitrified in the absence of Oxr1p. Here, we obtained two major classes, with the corresponding 3-D maps resolved at 7.3 and 7.4 Å resolution, respectively (Appendix Fig. S4B). For the Oxr1p free dataset, the two classes matched rotary states 2 (42%) and 1 (35%), respectively (Fig. 7B) (the remaining 23% of the particles did not align to a particular rotary state). In summary, the analysis shows that Oxr1p-bound holo V-ATPase is in rotary state 1, much like the $V_1$(C)Oxr1p complex isolated from the *vma13Δ* strain. Moreover, the observations that Oxr1p selectively binds (and probably disassembles) holoenzymes that are in rotary state 1 (Fig. 7A), and that approximately half of the wild-type complexes purified from vacuoles and vitrified in the presence or absence of Oxr1p are halted in state 2 (Fig. 7B; Appendix Fig. S4B) provide an explanation for why $V_1$ release from vacuoles is incomplete unless ATP is added to replenish the pool of state 1 enzymes.

## Discussion

We recently obtained a cryoEM structure of yeast $V_1ΔH$ bound to subunit C and Oxr1p, a protein that had not been seen associated with the enzyme previously (Khan et al, 2022). Subsequent biochemical experiments showed that Oxr1p caused disassembly of the holo V-ATPase into $V_1$ and $V_o$ subcomplexes in the absence of added ATP. Since it had been shown before that canonical in vivo V-ATPase regulation by reversible disassembly requires catalytically active enzyme (Liu and Kane, 1996; Parra and Kane, 1998), we speculated that Oxr1p may play a role in V-ATPase quality control by removing inactive (damaged) $V_1$ from the vacuolar membrane to allow for reassembly of the resulting $V_o$ complexes with undamaged $V_1$, thus maintaining a steady-state density of active pumps. However, when we tested this hypothesis at the onset of the current study, we found that Oxr1p is similarly efficient at disassembling inactive complexes (due to *CYS4* deletion or treatment with $H_2O_2$/ConA) as compared to active V-ATPases, indicating that Oxr1p does not have a preference for inactive over active complexes. While this finding did not rule out Oxr1p's involvement in V-ATPase quality control, it narrowed down Oxr1p's possible physiological function to playing a role in canonical reversible disassembly after all. To investigate this possibility, we generated *OXR1* deletion, rescue, and overexpression strains and found that none of the strains exhibited a Vma⁻ phenotype. Of note, whereas the *OXR1* deletion resulted in ~40% more $V_1$ on the vacuole, the ATPase activity remained similar to the wild-type control. Earlier work had shown that even wild-type vacuoles contain ~30% inactive (oxidized) V-ATPases (Oluwatosin and Kane, 1997), and it is possible that even more inactive complexes accumulate on vacuoles in the absence of Oxr1p. This could explain why the higher $V_1:V_o$ ratio on *oxr1Δ* vacuoles does not result in higher activity, supporting a role for Oxr1p in quality

control. On the other hand, vacuoles from the *OXR1* over-expression strain had not only less $V_1$, but also lower activity, suggesting an increased level of steady-state disassembly due to the excess of Oxr1p, an observation that offered further support for a role of Oxr1p in reversible disassembly.

Strikingly, when we followed the disassembly kinetics in the *oxr1Δ* strain using live-cell fluorescence microscopy of C-mNG, we found that V-ATPases no longer disassembled upon glucose withdrawal on a physiologically relevant timescale, a defect that could be rescued by expression of *OXR1* from a plasmid. This experiment, thus, provided the strongest evidence yet that Oxr1p is indeed responsible for efficient V-ATPase disassembly in vivo. And together with our earlier in vitro finding that Oxr1p causes disassembly of the purified enzyme, the live-cell imaging experiment supported a mechanism in which Oxr1p acted *directly* on the enzyme to exert its activity. When we followed the subunit C-mNG fusion for a longer time after glucose withdrawal, we found that after one hour, a subset of cells started to show some cytosolic fluorescence, indicating partial disassembly. Whereas the mechanism of this delayed release of $V_1$ from the vacuolar membrane is not known, we previously showed that ATP hydrolysis alone causes V-ATPase disassembly in vitro on a timescale similar to what we observed here in the *oxr1Δ* strain (Sharma and Wilkens, 2017). It is noteworthy that disassembly kinetics is also known to be linked to the intra- and extracellular pH (Dechant et al, 2010; Diakov and Kane, 2010), and V-ATPase mutants that do not affect activity but the rate of disassembly such as Glu44 in subunit E (Okamoto-Terry et al, 2013), Arg25 in subunit G (Charsky et al, 2000), and several residues in the non-homologous region (NHR) of the A subunit (Shao et al, 2003) have been described. Glu44 of E is in the Oxr1p binding site (Khan et al, 2022) (Appendix Fig. 4C), which could lower the affinity of V-ATPase for Oxr1p, thereby explaining the resistance of this mutant to disassembly. Arg25 in the G subunits of peripheral stalks EG2 and EG3 are close to the negatively charged C-termini of two of the A subunits, interactions that may be critical for the bending of these two peripheral stator stalks as seen in autoinhibited (Vasanthakumar et al, 2022) and Oxr1p-bound $V_1$ (Khan et al, 2022) (Appendix Fig. S4A). The mechanism by which the mutants in the NHR of A inhibit enzyme disassembly, however, is not obvious, except to say that the conformational changes occurring during disassembly may be impaired by the mutations at these locations.

Fluorescence microscopy showed that while some mNG-tagged Oxr1p localizes to mitochondria, most of the Oxr1p resides in the cytoplasm, consistent with Oxr1p's now established role in reversible disassembly. However, this observation is at odds with a prior report that placed Oxr1p exclusively into mitochondria (Elliott and Volkert, 2004), a discrepancy for which we have no explanation at the moment and resolving of which will require further study.

Whereas the *OXR1* deletion and rescue strains provided compelling evidence that Oxr1p is required for reversible disassembly in vivo, a question that remained was why the process in the cell can occur within a matter of minutes, unlike under in vitro conditions with purified components, where significant disassembly requires several hours? Previously, we showed that ATP hydrolysis greatly accelerates dissociation of $V_1$ subcomplexes from immobilized C subunit, in line with earlier studies that found that efficient release of $V_1$ from $V_o$ by chaotropic agents such as potassium nitrate required presence of ATP

(Parra and Kane, 1996; Sharma and Wilkens, 2017). Inspired by these earlier observations, we tested whether ATP could accelerate Oxr1p-mediated V-ATPase disassembly and found that not only did ATP accelerate disassembly of purified V-ATPase (reaching ~70% within 5 min, close to the physiological rate), active turnover also increased the extent of release of $V_1$ from vacuolar membranes. Taken together, this means that while Oxr1p can disassemble V-ATPase in absence of catalytic turnover, the free energy of ATP hydrolysis is required for the process to occur at a physiologically relevant rate. The finding also implies that it is the stability of the $V_1$-Oxr1p complex that provides the necessary energy for breaking the $V_1$–$V_o$ interface in absence of ATP.

We then asked what happens to Oxr1p following enzyme disassembly? We initially expected to be able to co-precipitate $V_1$ and Oxr1p from cleared lysate, but we found this not to be the case. Upon further investigation using purified wild-type vacuoles, we realized that Oxr1p-induced disassembly in absence of nucleotide produced a $V_1$ subcomplex containing Oxr1p and subunits H and C. Upon addition of ATP, this $V_1$(H,C)Oxr1p complex then released Oxr1p and the C subunit, but not H. This suggested that the $V_1$(H,C)Oxr1p complex that is generated upon treatment of V-ATPase with Oxr1p in absence of ATP represents a *disassembly intermediate*, and that subsequent ATP hydrolysis releases subunit C and Oxr1p (for it to catalyze additional rounds of disassembly) and enables the H subunit on the resulting $V_1$ subcomplex to adopt the autoinhibited conformation.

Previous structural studies of $V_1$ subcomplexes showed that while subunit H inhibited $V_1$-ATPase is mostly in state 2 (Oot et al, 2016; Vasanthakumar et al, 2022), Oxr1p-bound $V_1$ is found exclusively in state 1 (Khan et al, 2022). Since we find that Oxr1p-bound holoenzyme is also in state 1, this would suggest that Oxr1p-driven disassembly is *initiated* in state 1, and that the disassembly intermediate is also in state 1 as ATP is not required for its formation. In the presence of ATP, however, the disassembly intermediate is quickly converted to rotary state 2 characterized by a tight interaction between $H_{CT}$ and the N-termini of EG2 and trapping of inhibitory MgADP in one of the catalytic sites (Oot et al, 2016; Vasanthakumar et al, 2022). Whether the conversion from the state 1 intermediate to the state 2 autoinhibited conformation requires one or several ATP hydrolysis events is currently not known.

The binding site between Oxr1p and state 1 $V_1V_o$ or subunit C containing $V_1\Delta H$ ($V_1$(C)Oxr1p) is formed in part by the C-terminal domain of the B subunit that constitutes the open catalytic site (Appendix Fig. S4D). Upon conversion to state 2 via ATP hydrolysis, the B subunit C-terminal domain moves away from the Oxr1p binding site to bind the C-terminus of the neighboring A subunit to close the catalytic site, a conformational change that likely lowers the affinity for and facilitates release of Oxr1p (and with it, subunit C).

In summary, this indicates that the ATP hydrolysis that is required for efficient V-ATPase disassembly in vivo has several functions: (i) populating rotary state 1 for Oxr1p binding, (ii) overcoming activation barriers for breaking protein-protein interactions at the $V_1$–$V_o$ interface, (iii) trapping of inhibitory MgADP in one catalytic site, (iv) allowing $H_{CT}$ to assume its autoinhibitory conformation in state 2, and (v) freeing up of Oxr1p from the $V_1$(H,C)Oxr1p disassembly intermediate so that it can catalyze additional rounds of disassembly.

The conclusions from our in vivo and in vitro experiments together with the results from others lead to our current model for the molecular mechanism of V-ATPase regulation by reversible disassembly (Fig. 8). The model is divided into the three conditions *Glucose* (left circle), *No glucose* (middle circle), and *Glucose re-add* (right circle), the same conditions that were used to investigate Oxr1p's role in live cells. In the glucose condition (left circle), constitutively expressed Oxr1p binds V-ATPase in rotary state 1 causing disassembly, with subsequent ATP hydrolysis leading to release of Oxr1p and formation of autoinhibited $V_1$ in state 2 (Movie EV1). At this point, $V_o$ may still be in state 1, but since $c$-ring rotation does not require net free energy (no energy loss due to friction), the complex will quickly reach its energy minimum, rotational state 3, driven by thermal motion (Roh et al, 2020; Roh et al, 2018). Under the glucose condition, however, RAVE-catalyzed (re)assembly is dominant, resulting in a steady state of mostly assembled enzymes. Upon glucose withdrawal (middle circle), Oxr1p disassembles the enzyme at the same rate as in glucose, but since RAVE is no longer active, Oxr1p's action becomes dominant, resulting in a steady state of mostly disassembled enzymes. Upon restoring glucose (right circle), RAVE resumes activity, and the assembled state becomes dominant again.

While this manuscript was in preparation, a study was submitted to a preprint server (preprint: (Klössel et al, 2023)) in which Oxr1p and a second yeast TLDc protein called Rtc5p (**R**estriction of **t**elomere **c**apping protein 5) were characterized. The authors of the study found that deletion or overexpression of the two proteins, individually or in combination, did not produce a Vma⁻ phenotype, consistent with our observations for Oxr1p. They further showed that while individual deletions of *OXR1* or *RTC5* led to slightly more $V_1$ on vacuoles compared to wild type as measured using comparative proteomics, individual overexpression of the two proteins did not cause a significant reduction of assembled V-ATPases. This is in contrast to our observations showing significantly lower activity and reduced levels of $V_1$ for vacuoles purified from the *OXR1* overexpressed strain, a discrepancy which we cannot explain at the moment. Moreover, in agreement with our results, Klössel et al found that C-terminally mNG-tagged Oxr1p is localized to the cytosol, even though the construct appears to be non-functional. Since we find that C-terminally HA-tagged Oxr1p complements the *oxr1Δ* strain, it is possible that the significantly larger mNG (~27 kDa) prevents functional binding as Oxr1p's C-terminus is partially buried in the Oxr1p:$V_1$ interface (Khan et al, 2022). Thus, while our results with Oxr1p are largely consistent with Klössel et al's findings, more work is needed to elucidate Rtc5p's molecular role in V-ATPase regulation by reversible disassembly.

In higher eukaryotes, several homologs of yeast Oxr1p that share the conserved TLDc domain have been reported, including NCOA7, OXR1, TLDC2, TBC1D24 and mEAK7 (nomenclature of the human proteins), and while all of these have been shown to interact with the V-ATPase (Castroflorio et al, 2021; Eaton et al, 2021a; Merkulova et al, 2015; Tan et al, 2022; Wang et al, 2022a; Wang et al, 2022b), their effect on enzyme activity and regulation are only now beginning to be explored. However, based on the structural and functional similarities of the yeast and mammalian V-ATPases, and the conservation of the TLDc domain, it can be assumed that at least some of these proteins act directly on the mammalian enzyme in a fashion similar to what we have found for yeast Oxr1p. For example, subsequent to V-ATPase-driven loading of synaptic vesicles with neurotransmitters, the enzyme is known to disassemble in order to facilitate fusion of the filled vesicles with the

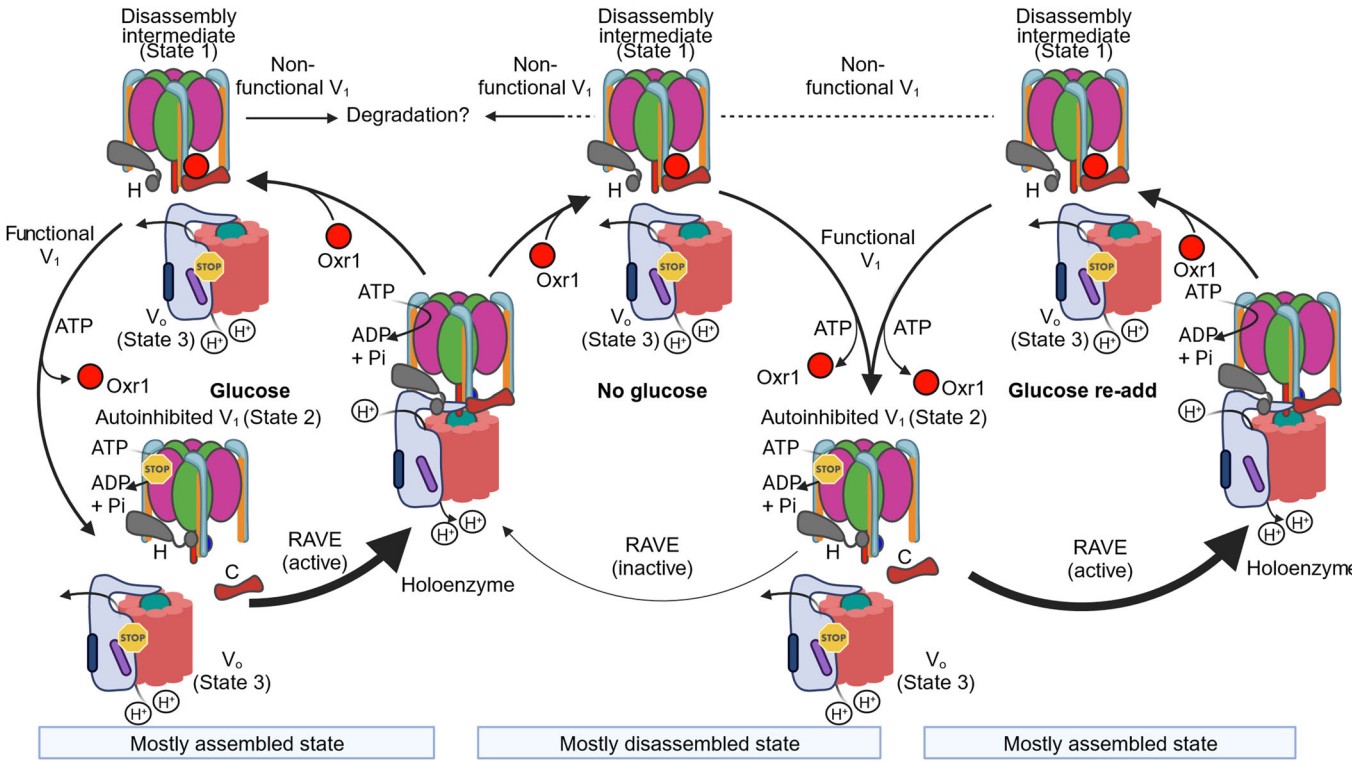

**Figure 8. Proposed mechanism of yeast V-ATPase regulation by reversible disassembly.**

Left circle: In the presence of glucose, the assembly chaperone RAVE is active and outcompetes the constitutive activity of Oxr1p and thus V-ATPase is mostly in the assembled state. Middle circle: In the absence of glucose, RAVE is cytosolic and inactive and as a result, Oxr1p's activity becomes dominant so that most of the enzymes are in the disassembled state. Right circle: After re-addition of glucose, RAVE is reactivated and restores the mostly assembled state. Of note, if the *disassembly intermediate* $V_1(C,H)Oxr1p$ originates from an enzyme that is inactive (damaged) due to e.g., oxidation of catalytic site cysteines, it may be targeted for degradation as part of $V_1$ quality control, or reactivated given such a mechanism exists. In addition, we have currently no experimental data to suggest how autoinhibited state 2 $V_1$ is reactivated and prepared for reassembly with state 3 $V_o$. See "Discussion" for additional details. The figure was created with BioRender.com.

presynaptic membrane (Bodzeta et al, 2017). Likewise, on lysosomes, V-ATPase has been shown to reversibly disassemble in response to nutrient levels under control of mTOR signaling (Ratto et al, 2022; Stransky and Forgac, 2015). The molecular mechanism by which V-ATPases on these membranes disassemble is not known, however, based on our findings with yeast Oxr1p, it is tempting to speculate that the process at these locations is facilitated by one or some of the mammalian TLDc proteins. This hypothesis is also supported by the observation that some of the pathological mutations in TLDc proteins phenocopy disease mutations in the V-ATPase (Merkulova et al, 2018). How mammalian TLDc proteins interact with V-ATPase and what the functional consequences of these interactions are opens up an entirely new field in V-ATPase research, insights from which may ultimately lead to new avenues for therapeutic intervention against diseases caused by V-ATPase malfunction.

## Methods

### Isolation and purification of yeast vacuoles

Yeast vacuoles were isolated and purified as described previously (Khan et al, 2022; Sharma and Wilkens, 2017). Briefly, yeast cells

were harvested at an OD of ~1.0, spheroplasted by zymolyase treatment and lysed with 10-12 strokes in a Dounce homogenizer in buffer A (10 mM MES-Tris pH 6.9, 0.1 mM $MgCl_2$, 12% Ficoll 400). Following ultracentrifugation (71,000 × g, 4 °C, 40 min), vacuoles were recovered from the top of the tube (floating white layer), resuspended in buffer A and overlaid with buffer B (10 mM MES-Tris pH 6.9, 0.1 mM $MgCl_2$, 8% Ficoll 400) and ultracentrifuged again. Vacuoles were again collected from the top layer and resuspended in 1.5 mM MES-Tris pH 7, 4.8% glycerol, and either used right away or stored at -80 °C. Of note, specific ATPase activities of purified vacuoles vary significantly depending on yeast strains (wild type with and without affinity tags, mutant strains) and the growth media (YEPD, synthetic complete (SC), synthetic dropout (SD)). Typical specific activities for wild-type in SC were 3.5–4.5 $\mu$mol × (min × mg)$^{-1}$; for wild-type in YEPD, 4–8 $\mu$mol × (min × mg)$^{-1}$; for wild-type with purification tag in YEPD, 1–2.5 $\mu$mol × (min × mg)$^{-1}$.

### Purification of wild-type V-ATPase and reconstitution into nanodisc

V-ATPase was purified and reconstituted into lipid nanodiscs as described (Sharma and Wilkens, 2017). Briefly, vacuoles from a yeast strain deleted for subunit G (SF838-5Aα-*vma10*Δ) and

complemented with pRS315 expressing N-terminally FLAG-tagged G were supplemented with protease inhibitors and solubilized in *n*-dodecyl β-d-maltopyranoside (DDM) at 1.2 mg/mg of vacuolar membrane protein for 1 h at 4 °C. Insoluble material was removed by ultracentrifugation (100,000 × *g*, 30 min) and the supernatant was mixed with biotinylated membrane scaffold protein (MSP1E3D1) (Sharma and Wilkens, 2017) at 1:50 molar ratio and rotated at 4 °C for 1 h followed by detergent removal with 0.4 g/ml Bio-beads SM2 for 2 h. The sample was then passed over 2 ml α-FLAG affinity resin, and the column washed with 10 column volumes (CV) of TBSE buffer (20 mM Tris pH 7.2, 150 mM NaCl, 0.5 mM EDTA) before eluting with 5 CV of 0.1 mg/ml FLAG peptide in TBSE. Fractions containing V-ATPase were identified by SDS-PAGE, pooled and concentrated using a 100 kDa MWCO centrifugal concentrator.

## Generation of yeast strains

All yeast strains were generated by transforming either plasmid DNA or PCR-amplified product using standard lithium-acetate procedures as described (Khan et al, 2022). Transformants were selected on synthetic dropout media or drug-containing YEPD plates as appropriate. In case of genomic integration, positive colonies were identified using colony PCR (Phire Plant Direct PCR Master Mix, ThermoScientific, catalog no. F160S), and the PCR products confirmed by DNA sequencing. For deletion of *OXR1*, a hygromycin cassette was amplified from a vector using Oxr1del FWD and Oxr1del REV primers and the PCR product was transformed into SF838-5Aα (Stevens et al, 1986) and BY4742-*vma10Δ-vma13Δ* (containing pRS315-FLAG-*VMA10*) (Diab et al, 2009) strains to generate SF838-5Aα-*oxr1Δ* (called *oxr1Δ*) and BY4742-*oxr1Δ-vma10Δ-vma13Δ* (pRS315-FLAG-*VMA10*) (called $V_1ΔH^{(ΔOxr1)}$) strains, respectively. To tag Oxr1p C-terminally with mNeonGreen (mNG), the mNG coding sequence was amplified using Gen-Oxr1CT-mNG F and Gen-Oxr1CT-mNG R primers and the PCR product was transformed into SF838-5Aα to generate strain SF838-5Aα-*OXR1*-mNG. To generate SF838-5Aα-*VMA5*-mNG and SF838-5Aα-*oxr1Δ-VMA5*-mNG strains, the mNG sequence was PCR-amplified using Scarletvma5CTF and mNeonvma5CTR primers and the PCR product was transformed into SF838-5Aα and SF838-5Aα-*oxr1Δ* strains, respectively. All other strains were generated by transformation with the respective plasmids: (1) *oxr1*<sup>NP</sup>: pRS316-*OXR1*-HA plasmid was transformed into *oxr1Δ*; (2) *oxr1*<sup>OE</sup>: YEp352-*OXR1*-HA plasmid was transformed into *oxr1Δ*; (3) SF838-5Aα-*oxr1Δ-VMA5*-mNG strain expressing Oxr1p (*oxr1*<sup>NP</sup>C-mNG): pRS316-*OXR1*-HA plasmid was transformed into SF838-5Aα-*oxr1Δ-VMA5*-mNG; (4) $V_1ΔH$ expressing C-terminally HA-tagged Oxr1p (*oxr1*<sup>NP</sup>*vma13Δ*): pRS316-*OXR1*-HA plasmid was transformed into $V_1ΔH^{(ΔOxr1)}$; (5) mNG-Oxr1p: pRS316-mNG-*OXR1* plasmid was transformed into *oxr1Δ*. Genotypes of yeast strains and primer sequences are provided in Appendix Tables S1 and S2, respectively.

## Plasmid construction

All plasmids were constructed using restriction-free cloning (Bond and Naus, 2012). A pair of hybrid primers were used to amplify the gene of interest and the amplified product was then used as a megaprimer to insert the gene into the target vector. The insertion

of the gene was then confirmed by restriction digestion and DNA sequencing. C-terminally HA-tagged Oxr1p was first generated in a pMECS vector in two steps. First, the Oxr1p coding sequence (excluding the stop codon) plus its 5′ UTR (333 nucleotides) was amplified from the yeast genome using Oxr1UTR_pMECS_FWD and Oxr1UTR_HA_pMECS_REV primers and placed at the N-terminus of the HA epitope coding sequence present in the pMECS vector, with the Oxr1p coding sequence and the HA tag separated by a linker peptide (SSAAA). Second, the 3′ UTR (333 nucleotides) of Oxr1p was amplified from the yeast genome using Oxr1-3UTR F and Oxr1_3UTR R primers and placed after the stop codon present at the C-terminus of HA to construct a pMECS-*OXR1*-HA plasmid. From the pMECS-*OXR1*-HA plasmid, *OXR1*-HA with or without *OXR1*'s UTRs were transferred to pRS316 and YEp352 (2 μ) vectors using Oxr1UTR_pRS316 F and Oxr1UTR_pRS316 R, and Oxr1_HA_YEp352_F and Oxr1_-HA_YEp352_R primers, respectively, to generate pRS316-*OXR1*-HA and YEp352-*OXR1*-HA vectors. The pRS316-mNG-*OXR1* plasmid was generated in two steps: first, plasmid pRS316-*OXR1*-HA was used to construct plasmid pRS316-mCherry-*OXR1*, and then mCherry was replaced with mNG. In plasmid pRS316-*OXR1*-HA, the Oxr1p sequence was replaced with mCherry using primers mCher_OxUTR_F and mCher_OxUTR_R, and then HA was replaced with Oxr1p sequence using primers Oxr1_mCherry_F and Oxr1_mCherry_R to generate plasmid pRS316-mCherry-*OXR1*. The mNG coding sequence was amplified using primers mCher_OxUTR_F and mNG_Oxr1_R, and the PCR-amplified product was then used as a megaprimer to replace mCherry in the vector to construct pRS316-mNG-*OXR1*. The pET28a-*OXR1*-HA and pET28a-BAP-*OXR1* (BAP= biotin acceptor peptide or Avi tag) plasmids were constructed using a modified pET28a vector (pET28a-BAP-MSP) containing a 7x His, Avi tag and Prescission protease cleavage site (Sharma and Wilkens, 2017). The *OXR1*-HA coding sequence was PCR-amplified from the pMECS-*OXR1*-HA vector using Oxr1_ETF and Oxr1_HA_ETR primers, and the PCR product was then used to replace BAP-MSP in the pET28a vector to construct the pET28a-*OXR1*-HA plasmid. To construct the pET28a-BAP-*OXR1* plasmid, Oxr1p's coding sequence was amplified from yeast genomic DNA using Oxr1_pET28a_F and Oxr1_pET28a_R primers and used as a megaprimer for insertion into the modified pET28a vector. Primer sequences are provided in Appendix Table S2.

## Purification of $V_1$ subcomplexes

Wild-type and mutant $V_1$ subcomplexes were purified as described (Khan et al, 2022). Briefly, cells expressing N-terminally FLAG-tagged subunit G were grown to ~3.5 OD in YEPD or synthetic dropout (SD-Leu) media, harvested and resuspended in TBSE buffer before storing at −80 °C for later use. Frozen cells were thawed, supplemented with 5 mM β-mercaptoethanol, protease inhibitors (leupeptin, pepstatin, and PMSF) and lysed by passing through a microfluidizer at 18,000 PSI. The lysate was cleared by two subsequent centrifugations at 4000 × *g* and 13,000 × *g*, respectively. The cleared lysate was then passed over 5 ml α-FLAG affinity resin pre-equilibrated with TBSE, washed in the same buffer, and eluted with FLAG peptide (0.1 mg/ml in TBSE). Fractions containing $V_1$ subcomplexes were then pooled, concentrated and applied to a Superose 6 Increase HiScale (16 mm × 400 mm) size-exclusion

chromatography (SEC) column attached to an ÄKTA FPLC. Fractions were analyzed by SDS-PAGE and $V_1$ containing fractions were pooled and concentrated using a 50 kDa MWCO centrifugal concentrator.

## V-ATPase disassembly from vacuoles

In total, 50–75 µg vacuolar protein was incubated with or without recombinant Oxr1p for 30 min at room temperature (a portion kept aside served as input on the blot). Vacuoles were then resuspended in buffer C (5 mM MES-Tris pH 6.9, 2.5 mM $MgCl_2$, 12.5 mM KCl) and ultracentrifuged at 37,000 × g for 20 min at 4 °C. The pellet fractions from each sample were resuspended in an equal amount of hot cracking buffer (8 M Urea, 5% SDS, 1 mM EDTA, 50 mM Tris-HCl pH 6.8, 10 mM DTT) (*pellet*). The supernatant was TCA precipitated and washed in cold acetone before resuspending in an equal amount of hot cracking buffer (*sup*). Samples were then heat treated (at 95 °C for 10 min for subunit A and Oxr1p, or at 65 °C for 15 min for subunits *a* and C) and separated by SDS-PAGE before analysis by western blot.

## Negative stain electron microscopy

Analysis of the assembly state of holo V-ATPase by negative stain EM was carried out as described (Khan et al, 2022). Briefly, 5 µl protein sample at a concentration of ~30 µg/ml from each of the four conditions ($V_1V_o$ only, $V_1V_o$ + Oxr1p, $V_1V_o$ + 4 mM MgATP, and $V_1V_o$ + Oxr1p + 4 mM MgATP) were applied to glow discharged carbon-coated copper grids. After 1 min of incubation, grids were briefly washed with water and then stained with 5 µl of 1% uranyl acetate solution for another minute. Micrographs were collected in SUNY Upstate Medical University's TEM core facility on a JEOL JEM-1400 transmission electron microscope equipped with Gatan Orius SC1000 CCD camera at ×200,000 magnification and 80 keV. Images were manually inspected and assembled and disassembled particles were counted from 20 micrographs for each condition.

## Collection and processing of cryoEM data

For cryoEM analysis, 2.5 µl protein samples at ~1–2 mg/ml (V-ATPase in lipid nanodisc with or without a twofold molar excess of Oxr1p) were vitrified on AuFlat 1.2/1.3 grids using a Leica EM GP2 with settings 6 °C, 85% humidity, 5 s blot time. Grids were mounted in a Gatan 914 side entry cryo holder and movies containing 25 frames (2 e⁻/frame) were recorded at 40,000× on a JEOL JEM-2100FS equipped with Gatan OneView and K2 Summit cameras using SerialEM (Schorb et al, 2019). Movies were combined using MotionCorr as implemented in Relion4 (Kimanius et al, 2021), and the CTF was corrected using ctffind-4.1.14 (Rohou and Grigorieff, 2015). Motion-corrected micrographs of the V-ATPase only sample were then imported into EMAN2 (Tang et al, 2007) and particles were picked in e2boxer.py using the neural net training option. Particle coordinates were then transferred to Relion4 and particle images were extracted as 480 × 480 pixel boxes (0.96 Å/pixel) and binned 3× to give 160 × 160 pixel boxes (2.88 Å/pixel). Extracted particles were subjected to several rounds of 2-D classification to remove bad particles. Good 2-D averages/classes were then used for training and particle picking in Topaz (Bepler et al, 2019) as

implemented in Relion4. Particles were extracted using a FOM of -3 and bad particles were removed by 2-D classification. The dataset was then 3-D classified using EMD-31538 (Khan et al, 2022) as a starting reference, and 3-D classes were refined using Refine3D. Particles from the images of the Oxr1p-containing sample were picked in Topaz using the training model obtained from the V-ATPase (no Oxr1p) dataset and analyzed using the same strategy. Maps and models were displayed in UCSF Chimera 1.17.3 and ChimeraX 1.6.1 (Pettersen et al, 2004) for image rendering.

## Pulldown of $V_1$ from vacuoles, cytosolic lysates and purified subcomplexes

### From vacuoles

Vacuoles were incubated with or without recombinant N-terminally 7×His tagged biotinylated Oxr1p (bio-Oxr1p) for 30 min at room temperature (*input* in the blot) and ultracentrifuged as described earlier. The pellet fractions were resuspended in hot cracking buffer (*pellet* in the blot). The supernatant from the Oxr1p treated sample was split and incubated with (a) buffer, (b) 4 mM MgATP and (c) 4 mM MgAMP-PNP for 30 min at 4 °C while mixing gently. An equal amount of supernatant from an untreated sample was incubated in buffer as control. Biotinylated Oxr1p was then captured via streptavidin beads for 60 min at 4 °C while rotating. The beads were separated (unbound proteins were loaded as *sup* in the blot) and washed 3 × with buffer by pelleting at 8000 × g for 1 min. To elute the protein, beads were heated at 95 °C for 10 min after adding hot cracking buffer (*elution* in the blot). Proteins were then heat treated (at 95 °C for 10 min for subunit A, H and Oxr1p, or at 65 °C for 15 min for subunit C) and analyzed by SDS-PAGE followed by western blot. In reverse pulldown experiment, vacuoles purified from $V_1$ΔG strain expressing FLAG-tagged G were mixed with recombinant Oxr1p, pulled down via anti-FLAG M2 affinity gel, and eluted with hot cracking buffer.

### From cytosolic lysate

Yeast strains deleted for *OXR1* were grown to the stationary phase. Around 50 $OD_{600}$ cells were harvested and stored at –80 °C. Cells were thawed, supplemented with protease inhibitors, and disrupted with glass beads for 10 min by vortexing with intermittent cooling every other minute. The cytosolic lysate was cleared by centrifugation at 16,000 × g for 10 min followed by ultracentrifugation at 100,000 × g for 30 min. Total protein concentration of the cytosolic lysate was measured using a modified BCA-TCA assay (Couoh-Cardel et al, 2015). Cytosolic lysate (~100 µg) was incubated with or without recombinant N-terminally 7×His tagged biotinylated Oxr1p for 30 min on ice. As a control, Oxr1p was incubated with buffer instead of lysate. Proteins were then pulled down via streptavidin beads and eluted as described in "From vacuoles", except that hot sample loading buffer was used instead of cracking buffer for elution.

### From purified $V_1$ subcomplexes

Purified $V_1$ subcomplexes (~10 µg; wild-type $V_1$ or $V_1$ΔH$^{(ΔOxr1)}$) with or without twofold molar excess of subunit C were mixed with recombinant biotinylated Oxr1p and incubated for 30 min at room temperature. Individual controls such as Oxr1p alone, Vma5p alone and Oxr1p + Vma5p were also included. Proteins were then pulled down and eluted as described in "From cytosolic lysate".

## Co-immunoprecipitation (Co-IP)

Co-immunoprecipitation of $V_1$ and Oxr1p was carried out as described (Jaskolka and Kane, 2020). Briefly, ~100 $OD_{600}$ of cells were harvested from an exponentially growing culture and stored at $-80\,°C$ until use. Cells were lysed using glass beads and cytosolic lysate collected by ultracentrifugation as described in "Pulldown". After measuring the protein concentration, 0.25–0.30 mg protein was TCA precipitated to serve as input, and 2.5–3.0 mg protein was used for Co-IP. The cytosolic lysate was first pre-cleared using protein A sepharose beads followed by incubation with either α-Vma1p (8B1F3 mAb) or α-HA antibody (ThermoFisher, catalog no. 26183) for 2 h at $4\,°C$ while mixing gently. The antibody–lysate mixture was then incubated overnight at $4\,°C$ while rotating in the presence of sepharose protein A beads. Centrifuged beads were then washed, and proteins were eluted using a hot cracking buffer at $95\,°C$ for 10 min. Eluted proteins were analyzed by SDS-PAGE followed by western blot.

## Preparation of whole-cell lysate

Whole-cell lysate was prepared as described (Zhang et al, 2011) with the following modifications. Briefly, exponentially growing yeast cells were harvested and stored at $-80\,°C$ until use. Cells were thawed, resuspended in 2 M lithium acetate, and incubated for 5 min on ice before centrifugation. Pellets were resuspended in 0.4 M NaOH, and incubated for 5 min on ice again. Pellets were then resuspended in hot cracking buffer and incubated at $95\,°C$ for 5 min. Cellular debris was pelleted by centrifugation at $16,000 \times g$ for 10 min, and supernatants containing whole-cell extracts were used immediately or stored at $-80\,°C$ until use.

## Biolayer interferometry (BLI)

BLI experiments were performed using an Octet RED384 system as described (Khan et al, 2022; Sharma and Wilkens, 2017). All experiments were carried out in black, flat-bottom 96-well plates with wells containing 200 µl buffer with or without protein samples at the specified concentrations. Before dipping into the sample wells, all biosensors were pre-equilibrated in buffer for at least 10 min. Experiments were carried out at $23\,°C$ with plates shaking at 1000 rpm, and measurements were taken at a rate of $5\,s^{-1}$. In all cases, individual controls for each condition were subtracted from the experimental data prior to analysis by the Octet Red HT data analysis v10 software.

### Subunit H binding to $V_1\Delta H$ subcomplex in the presence of Oxr1p
To assess the effect of Oxr1p on the interaction between $V_1\Delta H$ and subunit H, α-mouse IgG Fc capture biosensors (Octet AMC Biosensors; Sartorius/ForteBio catalog no.18-0025) were loaded with 1 µg/ml α-MBP (maltose binding protein) monoclonal antibodies (New England BioLabs, catalog no. E8032S). Antibody-coated biosensors were then dipped into wells containing 5 µg/ml N-terminal MBP fusions of $H_{WT}$ or $H_{chim}$. The MBP-$H_{WT}$ or -$H_{chim}$ loaded biosensors were then dipped into wells containing 40 nM $V_1\Delta H$ (association step) containing either (1) buffer (TBSE plus 0.5 mg/ml BSA), (2) Oxr1p (at equimolar concentration), (3) subunit C (twofold molar excess), and (4) Oxr1p + subunit C; for

the dissociation step, sensors were then dipped into buffer. Control experiments for non-specific binding of $V_1\Delta H$ + Oxr1p + subunit C to α-MBP antibody-coated biosensors were run in parallel and subtracted from the final plot.

### $V_1$ dissociation from wild-type $V_1V_oND$
Streptavidin-coated biosensors (Octet SA Biosensors; Sartorius/ForteBio catalog no.18-0009) were equilibrated in buffer (TBSE + 1 mM β-mercaptoethanol and 0.5 mg/ml BSA) to establish the initial BLI signal baseline. Biosensors were then loaded with lipid nanodisc reconstituted holo V-ATPase ($V_1V_oND$) at 10 µg/ml via biotinylated MSP using a predefined threshold level. After achieving $V_1V_oND$ binding, biosensors were rinsed in buffer (second baseline) and then dipped into wells for recording dissociation kinetics: (1) 1 mM MgATP, (2) 1 mM MgAMP-PNP, (3) 1 µM Oxr1p, (4) 1 mM MgATP + 1 µM Oxr1p, (5) 1 mM MgAMP-PNP + 1 µM Oxr1p, and (6) 1 mM MgATP + 300 mM $KNO_3$. Controls for non-specific binding (sensors without immobilized $V_1V_oND$) were run in parallel. Biosensors were then rinsed in buffer (third baseline) before dipping into wells containing 0.2 U/µl prescission protease (PPase) to cleave the MSP and release $V_oND$ (PPase cleavage site is located downstream of the biotinylation tag).

### Oxr1p binding to $V_1\Delta H^{(\Delta Oxr1)}$ subcomplex
Biotinylated Oxr1p (10 µg/ml) was immobilized on streptavidin biosensors to a predefined threshold level. Biosensors were rinsed in buffer (20 mM Tris-HCl pH 7.0, 0.5 mM EDTA, β-mercaptoethanol, 5.0 mg/ml BSA) before dipping into wells containing $V_1\Delta H^{(\Delta Oxr1)}$ (with or without recombinant subunit C) (association step). Biosensors were then moved to wells containing buffer to monitor the dissociation step. Individual controls for each condition were included for subtracting the signal due to non-specific binding of the protein to empty biosensors.

## ATPase assay

ATPase activity assays of vacuoles and purified V-ATPase were carried out as described (Khan et al, 2022). Briefly, 1 ml assays (50 mM HEPES pH 7.5, 25 mM KCl, 0.5 mM NADH, 2 mM phosphoenolpyruvate, 5 mM ATP, 30 units each of lactate dehydrogenase and pyruvate kinase) were prewarmed at $37\,°C$ for 10 min and supplemented with 4 mM $MgCl_2$. The reaction was initiated by adding 2.5–10 µg protein into the assay, and the drop in absorbance at 340 nm was monitored in a Varian Cary 100 Bio UV–Visible Spectrometer in kinetics mode. After establishing a linear rate of ATP hydrolysis, 200 nM of the specific V-ATPase inhibitor Concanamycin A (ConA) was added into the assay. The assay was maintained at $37\,°C$ using a temperature-controlled cuvette holder, and the rate of ATP hydrolysis was determined using the Kinetics Application as implemented in the Cary WinUV software package version 3.

## Expression and purification of recombinant proteins (Oxr1p, Oxr1p-HA, bio-Oxr1p, MBP-$H_{WT}$, MBP-$H_{chim}$, subunit C, and MSP)

Oxr1p, HA-tagged Oxr1p (Oxr1p-HA) and biotinylated Oxr1p (bio-Oxr1p) were expressed and purified as described for

Oxr1p (Khan et al, 2022). Briefly, *Escherichia coli* BL21DE3 cells expressing N-terminally 7×His tagged proteins were harvested, lysed, sonicated and cleared of the debris. Lysates were then passed over a Ni-NTA column before being further purified by a Superdex-75 SEC column (16 mm × 500 mm). MBP-H$_{WT}$, MBP-H$_{chim}$, and subunit C were expressed and purified using established protocols (Oot and Wilkens, 2010; Sharma et al, 2019; Sharma et al, 2018). Briefly, *E. coli* Rosetta2 cells expressing N-terminally MBP-tagged H$_{WT}$, H$_{chim}$ and C were harvested, lysed, sonicated, and cleared of the debris. MBP-tagged proteins were then captured by amylose resin and eluted in 10 mM maltose before being passed over a Superdex-75 SEC column (16 mm × 500 mm) to polish MBP-H$_{WT}$ and MBP-H$_{chim}$. For purification of subunit C, the MBP tag was cleaved using Prescission protease, and the sample was dialyzed to adjust the pH and salt concentration for subsequent removal of MBP using a diethylaminoethyl (DEAE) sepharose column. The flow-through of the column was dialyzed and concentrated before being further purified by a Superdex-75 SEC column (16 mm × 500 mm). Biotinylated membrane scaffold protein (MSP1E3D1) was expressed and purified from *E. coli* BL21DE3 cells as described (Sharma and Wilkens, 2017). Briefly, cells were grown in the presence of D-biotin, harvested, enzymatically lysed, and sonicated. Cellular debris and large organelles were removed by centrifugation and protein was captured by Ni-NTA resin. Eluted protein from the column was dialyzed, concentrated and stored at −80 °C until use.

## Western blot analysis

Western blot analysis was performed as described (Khan et al, 2022). Briefly, proteins were separated by SDS-PAGE and transferred to a low-fluorescence polyvinylidene fluoride (LF-PVDF) membrane. Blots were blocked for 1 h in TBST (20 mM Tris-HCl pH 7.5, 150 mM NaCl, 0.05% Tween-20) plus 5% milk before incubating overnight with primary antibodies at 4 °C with gentle agitation. After washing three times with TBST buffer, goat **α**-mouse IgG Alexa Fluor Plus 488 (ThermoFisher, catalog no. A32723) or goat **α**-rabbit IgG H&L (IRDye® 800CW) (Abcam, catalog no. ab216773) secondary antibody was added at a final dilution of 1:2000 or 1:10,000, respectively, and incubated for 60-90 min at room temperature. The blot was washed again, dried, and imaged in a Bio-Rad ChemiDoc MP Imaging System. All images were analyzed using the Bio-Rad Image Lab software. Primary antibodies used in this study include- mouse monoclonal IgG penta-His antibody (Qiagen, catalog no. 34660), mouse α-Vma1p (8B1F3), mouse α-Vph1p (10D7), mouse α-Vma5p (7A2), rabbit α-Vma13p (polyclonal), mouse α-HA (ThermoFisher, catalog no. 26183) and mouse α-GAPDH (ThermoFisher, catalog no. MA5-15738).

## Spot test

The growth phenotype of *OXR1* mutant strains was determined using a spot test as described (Khan et al, 2022). Briefly, yeast cells were grown to an OD$_{600}$ of ~1 in liquid synthetic complete (SC) or dropout (SD-URA) media. ~1 OD$_{600}$ cells were harvested, washed, and resuspended in 1 ml of sterile water. Cells were then serially diluted up to $10^{-4}$. In total, 6 µl from each dilution was spotted on YEPD, YEPD pH 5 and YEPD pH 7.5 + 60 mM CaCl$_2$ plates, and incubated at 25, 30, and 37 °C for 3–4 days before being imaged.

## Fluorescence microscopy

The localization of mNeonGreen (mNG) tagged Oxr1p and subunit C (Vma5p) in different growth conditions was carried out as described (Jaskolka and Kane, 2020). Briefly, cells were grown to exponential phase in SC or SD-URA media, stained with 8 µM FM4-64 (ThermoFisher, catalog no. T3166) in YEPD before recovery in SC or SD-URA media. Fluorescence images of cells were then collected using a Zeiss fluorescence microscope in DIC, TexasRed and GFP channels under (i) glucose-fed (+) condition, (ii) 15 min (30, 45, 60, 90, and 120 min for the time course study) after glucose starvation (−), and (iii) 15 min after glucose re-addition (−/+) to the starved cells. Images were then processed in Fiji ImageJ (Schindelin et al, 2012) and Adobe Photoshop. Pearson's coefficient was calculated using the Fiji ImageJ plugin "Just Another Colocalization Plugin" (JACoP) (Bolte and Corde-lieres, 2006). To determine the cellular localization of mNG-tagged Oxr1p, exponentially growing cells were stained with MitoTracker™ Red CMXRos dye (ThermoFisher, catalog no. M7512) for 10 min and washed twice with SC media before collecting fluorescence images using DIC, TexasRed, and GFP channels. Overlay images were created by stacking green and red images at 50% opacity over the DIC image, then brightness was increased to 50%.

## Data availability

The cryoEM maps of yeast V-ATPase with and without Oxr1p bound are deposited in the Electron Microscopy Data Bank under accession numbers EMD-42880 and EMD-42881.

## Peer review information

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

## Acknowledgements

The authors thank Dr. Patricia Kane for yeast strains *vma1Δ* and *cys4Δ* and for α-Vma1p, α-Vma13p, α-Vma5p and α-Vph1p antibodies, Dr. Brian Haarer for the YEp352 plasmid, and Drs. Rebecca Oot and Patricia Kane and Ms. Kassidy Knight for helpful discussions. This work was supported by NIH grant GM141908 to SW.

## Author contributions

**Md. Murad Khan**: Conceptualization; Formal analysis; Validation; Investigation; Visualization; Methodology; Writing—original draft; Writing—review and editing. **Stephan Wilkens**: Conceptualization; Resources; Formal analysis; Supervision; Funding acquisition; Validation; Investigation; Visualization; Methodology; Project administration; Writing—review and editing.

## Disclosure and competing interests statement

The authors declare no competing interests.

# Expanded View Figures

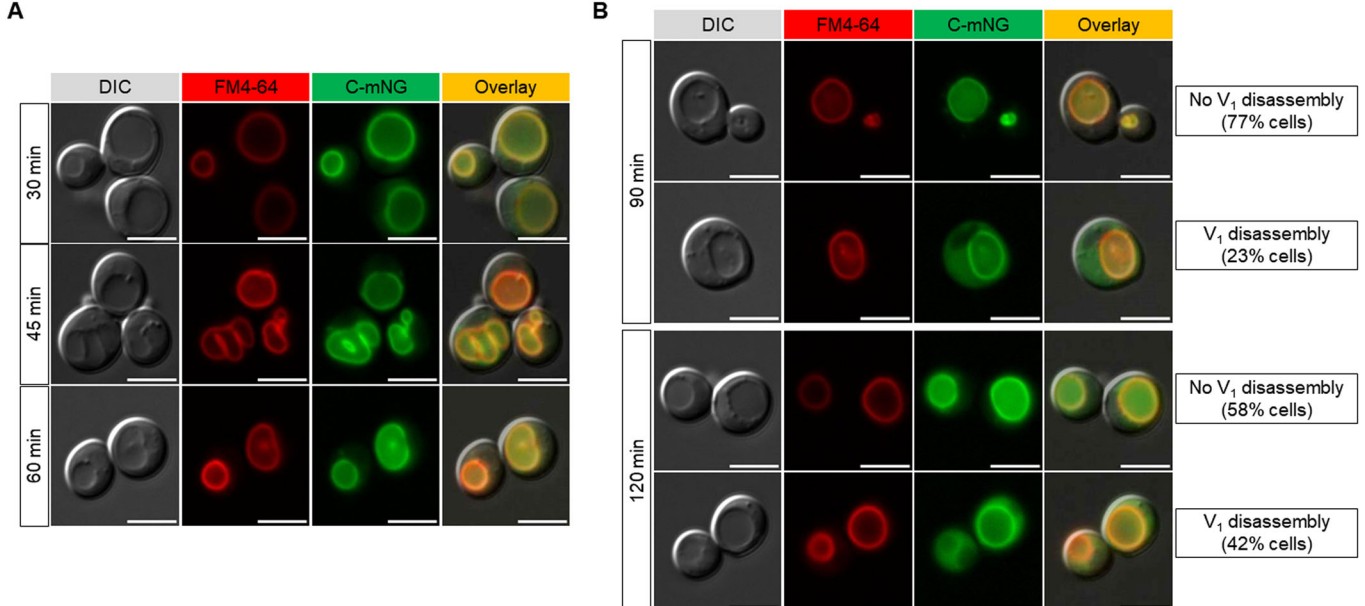

**Figure EV1. Time course analysis of V₁ disassembly in *oxr1Δ* yeast strain.**

(A, B) DIC and fluorescence images of *oxr1Δ* yeast cells expressing C-mNG after prolonged glucose deprivation (up to 2 h). Approximately 100 cells from two experiments were examined and cells showing diffuse staining (possibly indicating some form of V-ATPase disassembly) were expressed as a percentage of total cells at 1.5 and 2 h (B). Scale bar: 5 μm.

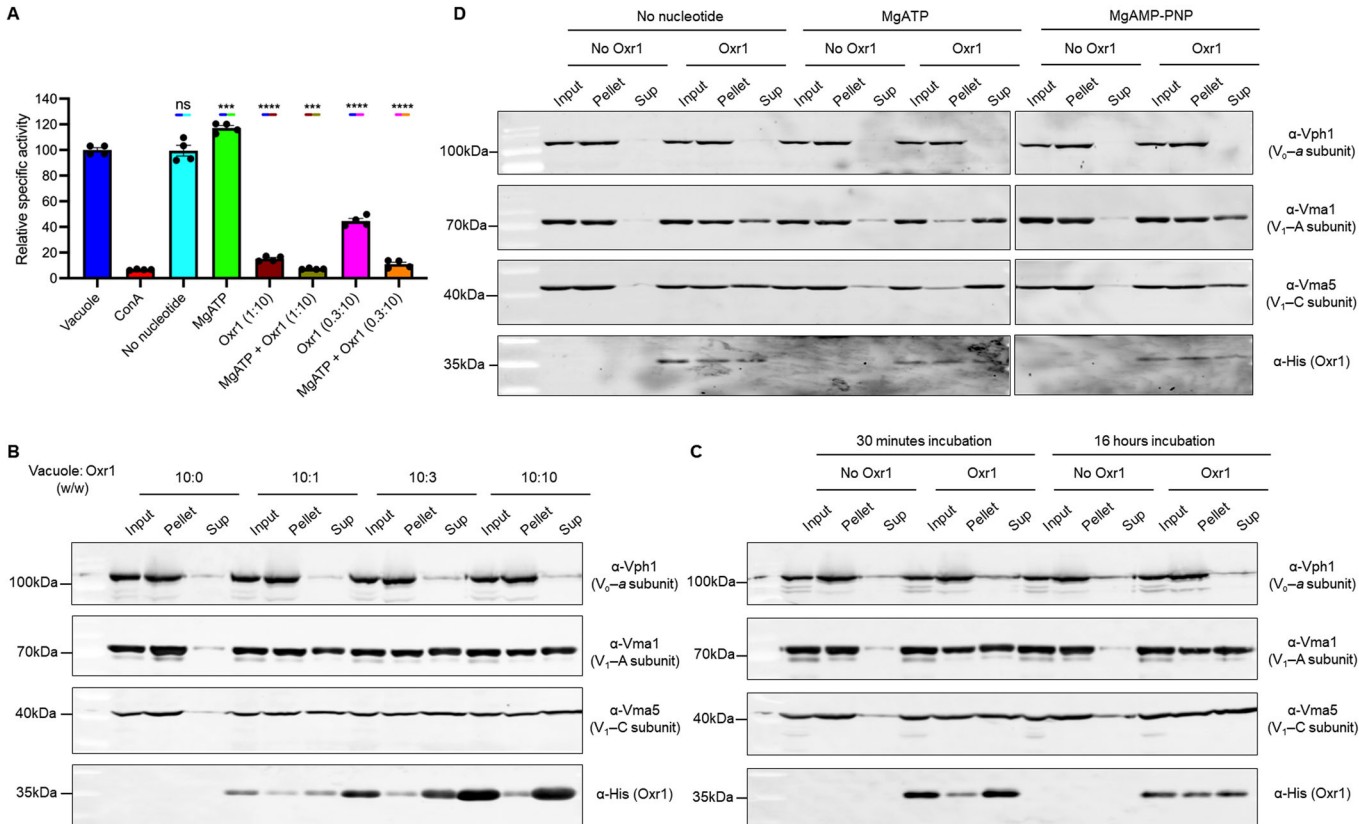

**Figure EV2. Increase efficiency of Oxr1p-mediated vacuolar ATPase activity inhibition and disassembly in presence of nucleotides.**

(A) Bar diagram of ATPase activities of purified vacuoles after incubating at room temperature for 30 min in absence and presence of Oxr1p (at 1:10 or 0.3:10 ratio (w/w) over total vacuolar protein) and 4 mM ATP. Relative activities were normalized against the starting activity of purified vacuoles (blue). ConA is used as an additional control (red). Individual data points of four tests from two biological preparations are shown. Data are presented as mean ± SEM. (B) Purified vacuoles were incubated with increasing concentrations of Oxr1p followed by ultracentrifugation. The extent of $V_1$ release from vacuoles was then determined by probing western blots of *pellet* and *sup* fractions with α-Vma1p ($V_1$ subunit A) and **α**-Vma5p ($V_1$ subunit C) antibodies. Increasing Oxr1p concentration does not lead to more $V_1$ disassembly from purified vacuoles, see equal amounts of Vma1p and Vma5p in the *sup* fraction in all Oxr1p treated samples. α-Vph1p ($V_0$ subunit *a*) and α-His (Oxr1p) blots were included as controls. A representative of three experiments from two biological preparations is shown. (C) The effect of longer incubation times on Oxr1p-mediated $V_1$ disassembly was determined by western blot analysis as described in (B). Increasing the incubation time of Oxr1p treatment does not lead to more $V_1$ disassembly from purified vacuoles, see the equal amounts of Vma1p and Vma5p in the *sup* fraction after both short and long incubation with a 1:10 ratio (w/w) of Oxr1p over vacuolar protein. α-Vph1p ($V_0$ subunit *a*) and α-His (Oxr1p) blots were included as controls. A representative of three experiments from two biological preparations is shown. (D) Purified vacuoles were incubated with Oxr1p (at 0.3:10 ratio (w/w) over vacuolar protein) in presence and absence of 4 mM nucleotides, and the extent of $V_1$ release from vacuoles was then determined by western blot analysis as described in (B). Relatively more intense bands of Vma1p and Vma5p in the ATP containing *sup* fraction indicates increased release of $V_1$ from $V_0$. α-Vph1p ($V_0$ subunit *a*) and α-His (Oxr1p) blots were included as controls. A representative of three experiments from two biological preparations is shown. Data information: Statistical significance (A) was calculated in GraphPad Prism 9 using unpaired Student's *t* test ($^{ns}$ indicates nonsignificant ($P > 0.05$); *** indicates $P \le 0.001$; **** indicates $P \le 0.0001$).

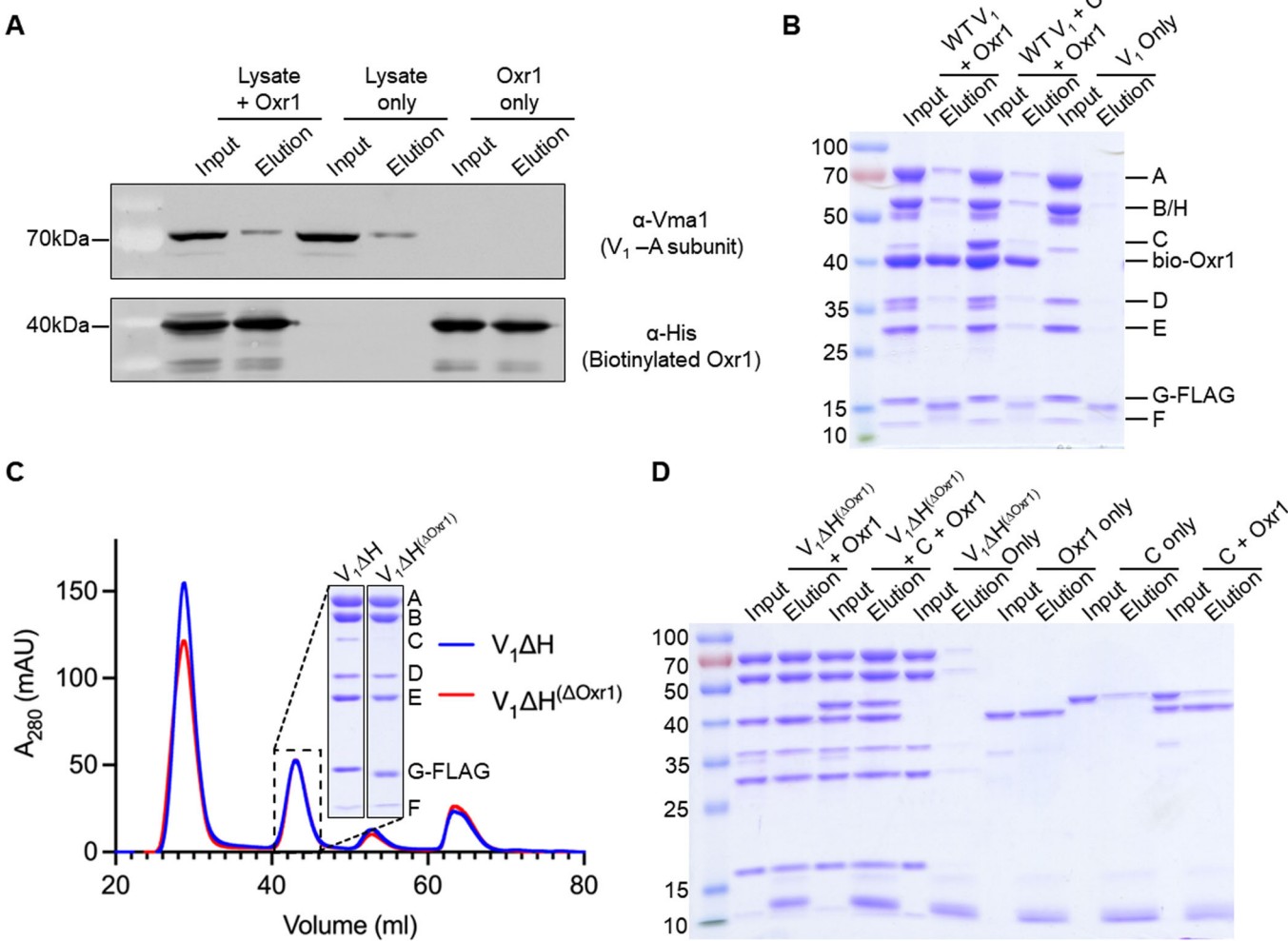

**Figure EV3. Oxr1p binding to wild-type and mutant V₁ subcomplexes.**

(A) Western blot of a pulldown of wild-type V₁ from cleared lysates of *oxr1Δ* cells using recombinant bio-Oxr1p as bait. The pulldown was carried out with lysate plus bio-Oxr1p, lysate plus buffer and bio-Oxr1p plus buffer. A representative of three experiments is shown. (B) Pulldown of purified wild-type V₁ in absence and presence of subunit C using recombinant bio-Oxr1p as bait as analyzed by SDS-PAGE. A representative of three experiments from two biological preparations is shown. (C) Size-exclusion chromatography (SEC) elution profile of V₁ΔH and V₁ΔH$^{(ΔOxr1)}$ (V₁ΔH purified from *oxr1Δ* background) on a Superose 6 increase HiScale column (16 mm × 400 mm). Inset, Coomassie blue stained SDS-PAGE gel of purified V₁ΔH and V₁ΔH$^{(ΔOxr1)}$. Two independent V₁ΔH$^{(ΔOxr1)}$ preparations gave an average specific activity of ~44 ± 8 μmol × (min × mg)$^{-1}$. (D) Pulldown of V₁ΔH$^{(ΔOxr1)}$ by recombinant bio-Oxr1p was performed under the indicated conditions and analyzed by Coomassie blue stained SDS-PAGE gel. The analysis suggests that bio-Oxr1p can efficiently pull down purified V₁ΔH$^{(ΔOxr1)}$ with or without subunit C. Labeling as shown in (B). A representative of three experiments from two biological preparations is shown. Note that avidin monomer in the *elution* fractions runs at ~15 kDa in SDS-PAGE gels, in between the bands of FLAG-tagged subunit G and subunit F (B, D).

