## [Peer Review File · EMBO Reports]

Molecular Mechanism of Oxr1p Mediated Disassembly of Yeast V-ATPase

Md. Murad Khan and Stephan Wilkens

Corresponding author(s): Stephan Wilkens (wilkenss@upstate.edu)

Review Timeline:

Submission Date:	3rd Dec 23
Editorial Decision:	2nd Jan 24
Revision Received:	2nd Feb 24
Editorial Decision:	6th Mar 24
Revision Received:	7th Mar 24
Accepted:	9th Mar 24

Transaction Report:

Dear Dr. Wilkens

Thank you for the submission of your research manuscript to our journal. We have now received the full set of referee reports that is copied below.

As you will see, the referees acknowledge that the findings are potentially interesting, but they also raise a number of concerns that need to be addressed.

Given these constructive comments, we would like to invite you to revise your manuscript with the understanding that the referee concerns (as detailed above and in their reports) must be fully addressed and their suggestions taken on board. Please address all referee concerns in a complete point-by-point response. Acceptance of the manuscript will depend on a positive outcome of a second round of review. It is EMBO Reports policy to allow a single round of revision only and acceptance or rejection of the manuscript will therefore depend on the completeness of your responses included in the next, final version of the manuscript.

We realize that it is difficult to revise to a specific deadline. In the interest of protecting the conceptual advance provided by the work, we recommend a revision within 3 months (April 2nd). Please discuss the revision progress ahead of this time with the editor if you require more time to complete the revisions.

I am also happy to discuss the revision further via e-mail or a video call, if you wish.

*****IMPORTANT NOTE:

We perform an initial quality control of all revised manuscripts before re-review. Your manuscript will FAIL this control and the handling will be delayed IN CASE the following APPLIES:

- 1) A data availability section providing access to data deposited in public databases is missing. If you have not deposited any data, please add a sentence to the data availability section that explains that.
- 2) Your manuscript contains statistics and error bars based on $n=2$. Please use scatter blots in these cases. No statistics should be calculated if $n=2$.

When submitting your revised manuscript, please carefully review the instructions that follow below. Failure to include requested items will delay the evaluation of your revision.*****

- 1) a .docx formatted version of the manuscript text (including legends for main figures, EV figures and tables). Please make sure that the changes are highlighted to be clearly visible.
- 2) individual production quality figure files as .eps, .tif, .jpg (one file per figure). Please download our Figure Preparation Guidelines (figure preparation pdf) from our Author Guidelines pages <https://www.embopress.org/page/journal/14693178/authorguide> for more info on how to prepare your figures.
- 3) a .docx formatted letter INCLUDING the reviewers' reports and your detailed point-by-point responses to their comments. As part of the EMBO Press transparent editorial process, the point-by-point response is part of the Review Process File (RPF), which will be published alongside your paper.
- 4) a complete author checklist, which you can download from our author guidelines (<<https://www.embopress.org/page/journal/14693178/authorguide>>). Please insert information in the checklist that is also reflected in the manuscript. The completed author checklist will also be part of the RPF.
- 5) Please note that all corresponding authors are required to supply an ORCID ID for their name upon submission of a revised manuscript (<<https://orcid.org/>>). Please find instructions on how to link your ORCID ID to your account in our manuscript tracking system in our Author guidelines (<<https://www.embopress.org/page/journal/14693178/authorguide#authorshipguidelines>>)
- 6) We replaced Supplementary Information with Expanded View (EV) Figures and Tables that are collapsible/expandable online. A maximum of 5 EV Figures can be typeset. EV Figures should be cited as 'Figure EV1, Figure EV2' etc... in the text and their respective legends should be included in the main text after the legends of regular figures.

7) Please note that a Data Availability section at the end of Materials and Methods is now mandatory. In case you have no data that requires deposition in a public database, please state so instead of refereeing to the database.

See also < <https://www.embopress.org/page/journal/14693178/authorguide#dataavailability>>. Please note that the Data Availability Section is restricted to new primary data that are part of this study.

Additional information on source data and instruction on how to label the files are available

<<https://www.embopress.org/page/journal/14693178/authorguide#sourcedata>>.

10) Figure legends and data quantification:

- the name of the statistical test used to generate error bars and P values,
- the number (n) of independent experiments (please specify technical or biological replicates) underlying each data point,
- the nature of the bars and error bars (s.d., s.e.m.)

- If the data are obtained from n {less than or equal to} 5, show the individual data points in addition to the SD or SEM.

- If the data are obtained from n {less than or equal to} 2, use scatter blots showing the individual data points.

11) Our journal encourages inclusion of *data citations in the reference list* to directly cite datasets that were re-used and obtained from public databases. Data citations in the article text are distinct from normal bibliographical citations and should directly link to the database records from which the data can be accessed. In the main text, data citations are formatted as follows: "Data ref: Smith et al, 2001" or "Data ref: NCBI Sequence Read Archive PRJNA342805, 2017". In the Reference list, data citations must be labeled with "[DATASET]". A data reference must provide the database name, accession number/identifiers and a resolvable link to the landing page from which the data can be accessed at the end of the reference. Further instructions are available at <<https://www.embopress.org/page/journal/14693178/authorguide#referencesformat>>.

12) All Materials and Methods need to be described in the main text. We would encourage you to use 'Structured Methods', our new Materials and Methods format. According to this format, the Materials and Methods section should include a Reagents and Tools Table (listing key reagents, experimental models, software and relevant equipment and including their sources and relevant identifiers) followed by a Methods and Protocols section in which we encourage the authors to describe their methods using a step-by-step protocol format with bullet points, to facilitate the adoption of the methodologies across labs.

More information on how to adhere to this format as well as downloadable templates (.doc or .xls) for the Reagents and Tools Table can be found in our author guidelines: <

<https://www.embopress.org/page/journal/14693178/authorguide#manuscriptpreparation>>. An example of a Method paper with Structured Methods can be found here: <<https://www.embopress.org/doi/10.15252/msb.20178071>>.

13) As part of the EMBO publication's Transparent Editorial Process, EMBO Reports publishes online a Review Process File to accompany accepted manuscripts. This File will be published in conjunction with your paper and will include the referee reports, your point-by-point response and all pertinent correspondence relating to the manuscript.

Yours sincerely,

Referee #1:

The authors deciphered novel mechanistic insights to explain how Oxr1p favors glucose-dependent disassembly of vacuolar ATPase (V-ATPase) in yeast. Oxr1p is a new factor that regulates V-ATPases and was recently discovered in the Wilken's laboratory. The work is comprehensive and data compelling, including cellular, biochemical and structural studies. Importantly the data support the conclusions summarized in the model presented. The work is significant and would be of interest to a broad scientific community, since V-ATPases are central players in many processes and diseases and both the V-ATPases and the TLDc family of proteins are highly conserved.

However, the text must be streamlined throughout the manuscript for clarity. This includes discrepancies between text and labeling in some figures. One of several examples is in Fig. EV3F, where the double deletion strain is labeled as V1, which readers will interpret as wt. Other example are the ratios in Fig. EV2 (10:1 in fig vs 1:10 in text).

Authors must clarify the experimental conditions for disassembly/reassembly in vivo. It is unclear if (i) glucose fed, (ii) glucose starved, and (iii) glucose restored are the growth conditions as stated for Fig. 3 or whether these are acute treatments as indicated somewhere else: "if we starve the cells of glucose longer than the 15 min used in our fluorescence microscopy assays"

Page 8: "consistent with the live cell microscopy data for wild type and Oxr1NP strains (Fig 4A and C)." Correction: Fig 3 A and C

Since ~25% disassembly is detected in the Ctrl WT (Fig 3A), which is very low as compared to other studies, including Dechant et al, 2010 which the authors used as reference (5-10 min time frame). In the present study, glucose deprivation more than one hour was required to detect 40% disassembly (EV1). Explain whether and how this low level of disassembly (Fig 3) would impact measurement of Oxr1p localization in vacuoles vs cytosol.

Tagged Oxr1p is abundant in the cytosol (Fig 4). However, it co-localizes with the mitochondria and vacuole, regardless of glucose condition. It appears that a population of Oxr1p resides on these membranes, which have been ruled out in the text. Explain.

Include Pearson's co-efficient plot for Fig 4A

Fig. EV3C,D. Include positive control for binding, if possible.

Fig 5b. What was the incubation time used?

Authors indicate that the V1-Oxr1-C intermediate is active, but they didn't measure ATPase activity in Fig 6. How does the time frame in Fig 6 compare to the time frame used to measure activity in fig 5A? What is the incubation time used in Fig 6?

Fig 6B. Label for subunit identification is blocking the V1 well. Fix.

Fig 6E. Clarify experimental conditions in the legend. Was the pull down done from membranes? Or Whole cell lysate? The fact that sup + pellet does not = input suggest that the gel wells were saturated for this Western blot and titration would be needed.

Explain why pellet is not included in samples with ATP and AMP-PNP. Explain why the Oxr1 samples (Sup) are different in Figs 6E and 6F?

Include the figure number on each figure to facilitate review.

Referee #2:

Vacuolar-type adenosine triphosphatases (V-ATPases) are large, membrane-embedded protein complexes that function as proton pumps in eukaryotic cells. V-ATPases are rotary proton pumps that serve as signaling hubs, including endosomes, lysosomes, and secretory vesicles. In 2022, Khan et al. determined the cryo-EM structure of a mutant V1 subcomplex bound to subunit C and Oxidation Resistance 1 (Oxr1). However, whether Oxr1-induced enzyme disassembly observed in vitro plays a role in canonical V-ATPase regulation by reversible disassembly in vivo, or a novel mechanism of V-ATPase quality control, is unknown. In this paper, Md. Murad Khan and Stephan Wilkens unravel the molecular mechanism of Oxr1-induced disassembly that occurs in vivo as part of the canonical V-ATPase regulation by reversible disassembly. The manuscript advanced our understanding of the disassembly of Yeast V-ATPase, but several questions remain unanswered. I believe the authors should be able to address those questions.

-Can the authors clarify the conformational changes between the disassembly intermediate (State 1) and autoinhibited V1 (State 2) via the cryo-EM structures? How did ATP hydrolysis-triggered conformational changes help the holoenzyme release subunit C and Oxr1? Do the cryo-EM structures provide any direct information?

-Does pRS316 belong to high-copy or low-copy number plasmid? What's the plasmid ORI? If pRS316 is a high-copy-number plasmid, the Oxr1NP is also over-expressed compared to wide-type of Oxr1. The authors may want to use other low-copy-number plasmid to repeat the experiments.

-For Fig 2C, whereas ATPase activities of Oxr1 Δ and wild type vacuoles were similar, those from Oxr1NP strain was reduced to ~75% of wild type. However, For Fig 2D, the relative V1 to Vo ratio for Oxr1NP has no significant difference to wide type, whereas Oxr1 Δ was significantly increased. Why?

-For Fig 2E, the error bar is too large. Please repeat more times or recalculate the band gray value because each experimental group has one outlier spot, respectively.

-What's the Pearson's co-efficient for Fig 4A?

-For Fig 5B, indicate the incubation time for Oxr1, MgATP, MgATP+Oxr1.

-For Fig 6D, indicate the Kd for V1 Δ H(Δ Oxr1). Compare V1 Δ H(Δ Oxr1) with V1 Δ H(Δ Oxr1)+C, and analysis the binding affinity in the corresponding context.

-In page 3, Oxr1p should stand for Oxidation Resistance protein 1. Please unify all the usage of Oxr1 and Oxr1p throughout this manuscript.

-Because the structures in Fig 1 is not new results, suggest authors change it to a Expanded View figure and reorder the sequences of all the figures.

-For Discussion, it not that accurately interpret the results, but repetitive with the Results section and it even contain a comprehensive review of the field to some degree. Please rephrase it, especially compact the content related to Figure 8.

Authors' response to Reviewers' comments:

We would like to thank reviewers for recognizing the significance of the work and for their overall positive evaluation of our manuscript titled "Molecular Mechanism of Oxr1p Mediated Disassembly of Yeast V-ATPase". In the following, we provide a point-by-point response to the comments and concerns raised by the reviewers, with reviewer comments in black and authors' responses in blue font. Please note that we have modified some of the main text (see track changes) in response to the reviewers' comments/suggestions and corrected a few minor errors/inconsistencies in the text and figures (e.g. adding scale bars in all microscopy images, using standard nomenclature (*OXR1*=gene, Oxr1p=protein, *oxr1Δ*= *OXR1* deletion strain) etc.).

Referee #1:

The authors deciphered novel mechanistic insights to explain how Oxr1p favors glucose-dependent disassembly of vacuolar ATPase (V-ATPase) in yeast. Oxr1p is a new factor that regulates V-ATPases and was recently discovered in the Wilken's laboratory. The work is comprehensive and data compelling, including cellular, biochemical and structural studies. Importantly the data support the conclusions summarized in the model presented. The work is significant and would be of interest to a broad scientific community, since V-ATPases are central players in many processes and diseases and both the V-ATPases and the TLDc family of proteins are highly conserved.

Authors' response:

We thank the reviewer for the overall positive evaluation of our manuscript.

Referee #1:

However, the text must be streamlined throughout the manuscript for clarity. This includes discrepancies between text and labeling in some figures. One of several examples is in Fig. EV3F, where the double deletion strain is labeled as V1, which readers will interpret as wt. Other example are the ratios in Fig. EV2 (10:1 in fig vs 1:10 in text).

Authors' response:

We thank the reviewer for pointing that out. We have now changed the label for "V₁" in Fig. EV3B and EV3F to "WT V₁" and "V₁ΔH^(ΔOxr1)", respectively, and we corrected the ratio of Oxr1p over total vacuolar protein in Fig. EV2. Please note that the original panel Fig. EV3F is now Fig. EV3D in the revised manuscript.

Referee #1:

Authors must clarify the experimental conditions for disassembly/reassembly in vivo. It is unclear if (i) glucose fed, (ii) glucose starved, and (iii) glucose restored are the growth conditions as stated for Fig. 3 or whether these are acute treatments as indicated somewhere else: "if we starve the cells of glucose longer than the 15 min used in our fluorescence microscopy assays"

Authors' response:

To analyze reversible disassembly in living cells, we followed the procedure given by Jaskolka & Kane, 2020 (cited in the Materials and Methods section). The standard conditions for imaging reversible disassembly are "glucose fed", 15 min after removal of glucose (glucose starved), and 15 min after glucose re-addition. The extended times post starvation (30, 45, 60, 90 and 120 min, as stated in the Materials and Methods section under "Fluorescence Microscopy"), were used to see whether disassembly will occur in the *oxr1Δ* strain upon

prolonged glucose starvation. To clarify these conditions, we have now added the following description to the legends of Figs. 3 and 4:

Legend of Fig. 3:

*"DIC and fluorescence images of cells expressing C-mNG (green) collected in glucose fed (+), **15 min** after glucose deprivation (-), and **15 min** after glucose re-addition (-/+ conditions from (A) WT, (B) *oxr1Δ*, and (C) *oxr1^{NP}* strains."*

Legend of Fig. 4B&C:

*"DIC and fluorescence images of yeast cells expressing C- (B) or N-terminally (C) mNG tagged Oxr1p recorded under glucose fed (+), **15 min** after glucose deprivation (-) and **15 min** after glucose re-addition (-/+ conditions (green)."*

Referee #1:

Page 8: "consistent with the live cell microscopy data for wild type and Oxr1NP strains (Fig 4A and C)."

Correction: Fig 3 A and C

Authors' response:

We thank the reviewer for pointing out the error, this has now been corrected.

Referee #1:

Since ~25% disassembly is detected in the Ctrl WT (Fig 3A), which is very low as compared to other studies, including Dechant et al, 2010 which the authors used as reference (5-10 min time frame).

Authors' response:

Again, we apologize for the lack of clarity. The experiment in Fig. 3 was not designed to obtain quantitative levels of disassembly, only **whether disassembly occurs, or not**. Consequently, we did not report a level of disassembly for any of the strains. We assume the "~25% disassembly" the reviewer refers to probably comes from looking at the Pearson's coefficient plot. However, the Pearson's coefficient in this case quantifies the overlap (or lack thereof) between the fluorescence signal of Vma5p-mNG and the vacuole specific dye, FM4-64. A low Pearson's coefficient in glucose deprived conditions as compared to glucose fed and/or glucose re-add conditions indicate that there is less V₁ subcomplex on the vacuole due to V-ATPase disassembly.

Referee #1:

In the present study, glucose deprivation more than one hour was required to detect 40% disassembly (EV1). Explain whether and how this low level of disassembly (Fig 3) would impact measurement of Oxr1p localization in vacuoles vs cytosol.

Authors' response:

Here we would like to clarify that in Fig. EV1, we measured the disassembly of V-ATPase over extended times in the **oxr1Δ strain** (not in the WT yeast strain). The experiment says that until one hour post glucose starvation, we did not observe any *oxr1Δ* cells that showed disassembly of the V-ATPase. However, upon longer observation (>1 h), we started to observe disassembly in a subset of cells, up to ~40% of the cells after two hours. To reiterate, the "~40%" is the percentage of total cells displaying disassembly, not the extent of V-ATPase disassembly in each cell.

Referee #1:

Tagged Oxr1p is abundant in the cytosol (Fig 4). However, it co-localizes with the mitochondria and vacuole, regardless of glucose condition. It appears that a population of Oxr1p resides on these membranes, which have been ruled out in the text. Explain.

Authors' response:

Based on our microscopy data (Fig. 4), it appeared that some Oxr1p may reside within mitochondria, and we were careful to point this out. Moreover, we do of course agree with the reviewer that Oxr1p does co-localize with the vacuolar membrane (as indicated by the Pearson's coefficient in Fig. 4B,C), which is consistent with our finding that Oxr1p binds the V-ATPase on the vacuolar surface prior to enzyme disassembly.

In the Results section of the original manuscript under "*Oxr1p is localized to the cytoplasm*", we concluded:

*"Moreover, the evidence presented above that Oxr1p is directly involved in reversible disassembly at the level of the enzyme means that **Oxr1p must be able to access the vacuolar membrane** under conditions that promote V-ATPase disassembly."*

And:

*"The fluorescence microscopy images show diffuse mNG fluorescence throughout the cytoplasm for both strains, with no obvious mNG fluorescence localization to mitochondria, which we visualized using MitoTracker dye (Fig 4A). **Whereas this experiment does not exclude the possibility that some Oxr1p is targeted to mitochondria**, the images show that the bulk of mNG tagged Oxr1p is cytosolic, consistent with Oxr1p's ability to bind and disassemble the V-ATPase in vivo."*

See also our response to the reviewer's next comment.

Referee #1:

Include Pearson's co-efficient plot for Fig 4A

Authors' response:

We thank the reviewer for the suggestion. We now have determined the Pearson's coefficient for the overlap of the fluorescence signals from Oxr1p-mNG and MitoTracker dye. The analysis indicates that there is a significant degree of overlap, suggesting that some of the Oxr1p-mNG may indeed be localized to mitochondria. We have now modified the main text in the Results and Discussion section as follows:

Page 8:

"The fluorescence microscopy images show diffuse mNG fluorescence throughout the cytoplasm for both strains, with some mNG fluorescence co-localizing to MitoTracker dye fluorescence as quantified using the Pearson's coefficient (Fig 4A). Thus, whereas the experiment suggests that some Oxr1p may be targeted to mitochondria, the images show that the bulk of mNG tagged Oxr1p is cytosolic, consistent with Oxr1p's ability to bind and disassemble the V-ATPase on the vacuolar membrane in vivo."

Page 18-19:

"Fluorescence microscopy showed that while some mNG tagged Oxr1p is localized to mitochondria, most of the Oxr1p resides in the cytoplasm, consistent with Oxr1p's now established role in reversible disassembly. However, this observation is at odds with a prior report that placed Oxr1p exclusively into mitochondria (Elliott et al, 2004), a discrepancy for which we have no explanation at the moment and resolving of which will require further study."

Referee #1:

Fig. EV3C,D. Include positive control for binding, if possible.

Authors response

We have performed these experiments three times each, with consistent results, so we are confident that our conclusion that wild type V_1 does not bind Oxr1p with high affinity is valid. The BLI based conclusion is also supported by the pulldown experiments presented in Figs. 6B and EV3A,B that indicate a lack of binding between Oxr1p and wild type V_1 . Moreover, we performed multiple BLI experiments using the same setup with V_1 purified from the *vma13Δ* strain ($V_1\Delta H$) instead of wild type V_1 and saw consistent binding to bio-Oxr1p (in a subunit C dependent manner). We of course appreciate the reviewer's point that a positive control would increase confidence in the conclusion reached from the BLI experiment, considering that panels EV3C,D present negative results. On the other hand, considering that the information content of these negative results is limited (and redundant with the conclusions of Figs. 6B and EV3A,B), and that repeating the experiment would likely require more time than allocated for the revision (multiple preparations of wild type and $V_1\Delta H$ would be needed, each taking 1-2 weeks), we have decided to remove panels C and D from Fig. EV3. Note that panels E and F in the original Fig. EV3 are now panels C and D in the revised Fig. EV3.

Referee #1:

Fig 5b. What was the incubation time used?

Authors' response:

As mentioned in the legend of the figure, the incubation time of the experiment in Fig. 5B was 30 min (Legend of Fig 5B: *ATPase activities of purified V-ATPase pre-incubated for **30 min** with a 2-fold molar excess of Oxr1p in absence (cyan) or presence (magenta) of 4 mM MgATP. As a control, ATPase activity was measured in presence of 4 mM MgATP only (green).*

This has now been clarified in the main text as well:

Page 9-10:

*"We therefore asked whether ATP might affect the rate of Oxr1p mediated disassembly. To address this question, we incubated wild type V-ATPase with a two-fold molar excess of Oxr1p or ATP individually or in combination for **30 min**, and measured ATPase activities. Whereas Oxr1p alone resulted in ~30% loss of activity, inhibition was close to complete in presence of Oxr1p and ATP (**Fig 5B**), with negative stain EM confirming virtually complete V-ATPase disassembly (**Fig 5C and D**)."*

Referee #1:

Authors indicate that the V_1 -Oxr1-C intermediate is active, but they didn't measure ATPase activity in Fig 6.

Authors' response:

We thank the reviewer for pointing this out. Our conclusion that the disassembly intermediate ($V_1(H,C)Oxr1p$) has transient ATPase activity is based on the fact that both Oxr1p and subunit C get released from the disassembly intermediate in presence of ATP, but not in presence of the non-hydrolyzable analog, AMP-PNP, indicating that hydrolysis of ATP is required. This ATP hydrolysis by the disassembly intermediate leads to the formation of autoinhibited V_1 . We cannot say how many molecules of ATP need to be hydrolyzed for Oxr1p and C to get released from the disassembly intermediate and for the V_1 to adopt the autoinhibitory conformation. However, to address the reviewer's comment, we have now performed the experiment and added the disassembly intermediate released from wild type vacuoles by the addition of Oxr1p to our standard ATP regenerating assay. However, the resulting trace resembles the trace of autoinhibited wild type V_1 , indicating that the disassembly intermediate is converted to the autoinhibited conformation after only a few turnovers, too few to be observed in the assay. We have included the results of these experiments in Fig. R1 below (only for review purposes):

Figure R1: Representative ATPase activity assay of the disassembly intermediate.

(A) 100 μ l of purified vacuoles (~ 1 mg/ml vacuolar protein, $\sim 5 \mu\text{mol} \times (\text{min} \times \text{mg})^{-1}$) were incubated with or without Oxr1p (at a 1:10 ratio (w/w) of Oxr1p over total vacuolar protein) for 30 min at room temperature and activities after incubation with buffer (blue trace) and Oxr1p (cyan trace) were measured in an ATP regenerating system. Under these conditions, inhibition was approximately 75%. (B) Purified vacuoles pre-incubated with or without Oxr1p were then ultracentrifuged. Supernatants from the ultracentrifugation step were analyzed by SDS-PAGE and activities were measured. Note that the assay in (B) was supplemented with concanamycin A (ConA) before adding supernatant in order to inhibit any residual activity from contaminating holoenzyme (if present). Also note that ConA inhibits the holoenzyme by binding to the V_o proton channel and that ConA does not inhibit ATPase activity of isolated active mutant V_1 complexes. Activities of disassembly intermediate released from vacuoles upon Oxr1p treatment were measured five times using two separate biological vacuole preparations, with all experiments giving comparable results.

Referee #1:

How does the time frame in Fig 6 compare to the time frame used to measure activity in fig 5A? What is the incubation time used in Fig 6?

In Fig. 5A, we have measured the activity of purified V-ATPase in presence of Oxr1 over time (5 min, 30 min, 16 hours). However, as in Fig 5B and Fig EV2D, we have shown that complete inhibition of ATPase activity (of purified V-ATPase and purified vacuoles) in presence of ATP does not require more than 30 min. In the experiment in Fig 6E and F (we assume the reviewer was referring to panels E and F of Fig 6), we have incubated the samples (Oxr1p + vacuoles and resulting disassembly intermediate) for 30 min each. This is explained in the Materials and Methods section under "Pull-down of V_1 from vacuoles, cytosolic lysates and purified subcomplexes":

"From vacuoles — Vacuoles were incubated with or without recombinant N-terminally 7 \times His tagged biotinylated Oxr1p (bio-Oxr1p) for **30 min at room temperature** (input in the blot) and ultracentrifuged as described earlier. The pellet fractions were resuspended in hot cracking buffer (pellet in the blot). The supernatant from the Oxr1p treated sample was split and incubated with (a) buffer, (b) 4 mM MgATP and (c) 4 mM MgAMP-PNP for **30 min at 4 $^{\circ}$ C** while mixing gently. An equal amount of supernatant from an untreated sample was incubated in buffer as control. Biotinylated Oxr1p was then captured via streptavidin beads for **60 min at 4 $^{\circ}$ C** while rotating"

We have added the following text to the legend for clarification:

Legend of Fig. 6E:

"Western blot of a pull-down of disassembled V₁ complexes in absence or presence of the indicated nucleotides using bio-Oxr1p as bait. Purified vacuoles were incubated with or without bio-Oxr1p for 30 min before ultracentrifugation. Supernatants were then incubated either with buffer or nucleotides for an additional 30 min before bio-Oxr1p pull down via streptavidin beads for 1 h (see Methods section for details)."

Referee #1:

Fig 6B. Label for subunit identification is blocking the V1 well. Fix.

Authors' response:

We assume the reviewer meant Fig. EV3B instead of Fig. 6B? We now shortened the lines pointing to subunit bands in Fig. EV3B and we thank the reviewer for pointing this out.

Referee #1:

Fig 6E. Clarify experimental conditions in the legend. Was the pull down done from membranes? Or Whole cell lysate?

Authors' response:

The pull-down in Fig. 6E was carried out from purified vacuoles. This has now been stated in the revised legend to Fig. 6E.

Legend of Fig 6E:

"Western blot of a pull-down of disassembled V₁ complexes in absence or presence of the indicated nucleotides using bio-Oxr1p as bait. Purified vacuoles were incubated with or without bio-Oxr1p for 30 min before ultracentrifugation."

Referee #1:

The fact that sup + pellet does not = input suggest that the gel wells were saturated for this Western blot and titration would be needed.

The reviewer is correct to say that the sum of "sup" + "pellet" does not equal the "input". This is because the amount of "input" loaded was different from the amounts of "sup" and/or "pellet", but not because of saturation of the blot. In the western blot (WB), we used a fluorophore tagged secondary antibody that is more resistant to saturation as compared to chemiluminescence. Fluorophore based WBs give band intensities that are proportional to the amount of the protein present in the wells. However, the amount of "pellet", "sup" and "elution" that was loaded onto the gel was equal and comparable in all conditions, and the amount of "input" that was loaded in different conditions was equal to each other, but not equal to "pellet", "sup" or "elution". The "input" was loaded only to show that we started with comparable amounts of vacuoles in all conditions, and that all subunits that we wanted to detect were detectable under the experimental conditions. Moreover, in the experiment, we just compared the relative presence of V₁ subunits and Oxr1p in "sup" and "elution" fractions that were loaded in equal amounts across the conditions.

Referee #1:

Explain why pellet is not included in samples with ATP and AMP-PNP.

Authors' response:

Purified vacuoles were incubated with Oxr1p and ultracentrifuged before the supernatant was split into three groups (Oxr1p, Oxr1p + ATP, and Oxr1p + AMP-PNP). This means that all three samples have the same "input" and "pellet". To simplify the blot, "input" and "pellet" fractions were only shown once rather than in triplicate. This is explained in the Materials and Methods section under "*Pulldown of V1 from vacuoles, cytosolic lysates and purified subcomplexes*":

*"From vacuoles — Vacuoles were incubated with or without recombinant N-terminally 7×His tagged biotinylated Oxr1p (bio-Oxr1p) for 30 min at room temperature (input in the blot) and ultracentrifuged as described earlier. The pellet fractions were resuspended in hot cracking buffer (pellet in the blot). **The supernatant from the Oxr1p treated sample was split and incubated with (a) buffer, (b) 4 mM MgATP and (c) 4 mM MgAMP-PNP** for 30 min at 4 °C while mixing gently. An equal amount of supernatant from an untreated sample was incubated in buffer as control. Biotinylated Oxr1p was then captured via streptavidin beads for 60 min at 4 °C while rotating."*

Referee #1:

Explain why the Oxr1 samples (Sup) are different in Figs 6E and 6F?

Authors' response:

In Fig 6E and 6F, we have used two different forms of Oxr1p. In Fig 6E, Oxr1p has been biotinylated (bio-Oxr1p) via an N-terminal "Bap" (or "Avi") tag, but in Fig. 6F, Oxr1p does not have the Avi Tag. This is why the Avi tagged Oxr1p has slightly higher MW. In Fig. 6E, we carried out the pulldown using bio-Oxr1p as bait. However, biotinylation of Oxr1p is not always complete. As a result, streptavidin beads could not pull down some Avi tagged Oxr1p (although the Avi tag is present, the tag is not biotinylated) present in the sample (both free and V₁ bound). That's why we see the bands of V₁ subunits in the "sup" fraction of the Oxr1p sample in Fig 6E. On the other hand, in Fig 6F, pulldown was performed using the FLAG-tagged G subunit of the V₁ subcomplex as bait, which has been efficiently pulled down via α-FLAG beads. As a result, we don't see significant bands for V₁ subunits A and C in the sup fraction, except for some minor amounts of subunit H (which gets released in presence of Oxr1p upon longer incubation times as mentioned in the legend) and free Oxr1p. To clarify this, we have included the following sentence in the legend to Fig. 6E:

Legend to Fig. 6E:

"Note that due to the incomplete biotinylation of Avi tagged Oxr1p (bio-Oxr1p), the sup fractions of bio-Oxr1p treated conditions have bands for bio-Oxr1p and V₁ subunits that are not pulled down by streptavidin beads."

Referee #1:

Include the figure number on each figure to facilitate review.

Authors' response:

We apologize for the inconvenience. Annotating figures with figure numbers would make that annotation part of the final figure as figure files are uploaded separately from the manuscript text during which time, we provide figure numbers for each uploaded figure file. The Journal then combines figures and manuscript text to generate a combined manuscript file in pdf format. We have no control over that process.

Referee #2:

Vacuolar-type adenosine triphosphatases (V-ATPases) are large, membrane-embedded protein complexes that function as proton pumps in eukaryotic cells. V-ATPases are rotary proton pumps that serve as signaling hubs, including endosomes, lysosomes, and secretory vesicles. In 2022, Khan et al. determined the cryo-EM structure of a mutant V1 subcomplex bound to subunit C and Oxidation Resistance 1 (Oxr1). However, whether Oxr1-

induced enzyme disassembly observed in vitro plays a role in canonical V-ATPase regulation by reversible disassembly in vivo, or a novel mechanism of V-ATPase quality control, is unknown. In this paper, Md. Murad Khan and Stephan Wilkens unravel the molecular mechanism of Oxr1-induced disassembly that occurs in vivo as part of the canonical V-ATPase regulation by reversible disassembly. The manuscript advanced our understanding of the disassembly of Yeast V-ATPase, but several questions remain unanswered. I believe the authors should be able to address those questions.

Authors' response:

We thank the reviewer for the overall positive evaluation of our manuscript.

Referee #2:

-Can the authors clarify the conformational changes between the disassembly intermediate (State 1) and autoinhibited V₁ (State 2) via the cryo-EM structures?

Authors' response:

We currently do not have a structure of the disassembly intermediate. We do have a low-resolution (~8 Å) cryoEM map of the upstream complex of the holoenzyme (V₁V_o) bound to Oxr1p in rotary state 1 (Fig. 7) and we have our structure of the V₁(C)Oxr1p complex in state 1 (Fig. 1B; pdb id: 7fde) that was isolated from a strain deleted for subunit H (*vma13Δ*). In addition, there is the cryoEM structure of autoinhibited V₁ in state 2 from the Rubinstein lab (pdb id: 7tmm). Based on these structures, we generated Movie EV1 that morphs conformations starting from binding of Oxr1p to state 1 V₁V_o - going to state 1 V₁(C)Oxr1p (7fde) - and then to autoinhibited state 2 V₁ (7tmm). This mechanism is now described in more detail in the Discussion section:

Page 20:

"Previous structural studies of V₁ subcomplexes showed that while subunit H inhibited V₁-ATPase is mostly in state 2 (Oot et al, 2016; Vasanthakumar et al, 2022), Oxr1p bound V₁ is found exclusively in state 1 (Khan et al, 2022). Since we find that Oxr1p bound holoenzyme is also in state 1, this would suggest that Oxr1p driven disassembly is initiated in state 1, and that the disassembly intermediate is also in state 1 as ATP is not required for its formation. In presence of ATP, however, the disassembly intermediate is quickly converted to rotary state 2 characterized by a tight interaction between H_{CT} and the N-termini of EG2 and trapping of inhibitory MgADP in one of the catalytic sites (Oot et al, 2016; Vasanthakumar et al, 2022). Whether the conversion from the state 1 intermediate to the state 2 autoinhibited conformation requires one or several ATP hydrolysis events is currently not known."

Referee #2:

How did ATP hydrolysis-triggered conformational changes help the holoenzyme release subunit C and Oxr1? Do the cryo-EM structures provide any direct information?

Authors' response:

In rotary state 1, Oxr1p is bound to the C-terminal domain of the open catalytic site as well as EG2 and C (this is true for both V₁ and V₁V_o; see revised Appendix Fig. S4D). Upon ATP hydrolysis, the C-terminus of the B subunit retracts to interact more closely with the C-terminus of the A subunit, leading to this catalytic site now being closed. We think that the conformational change of the B subunit C-terminal domain lowers the affinity for Oxr1p (and with it, subunit C). Subunit C binding is further destabilized by bending of the N-terminal segments of peripheral stator EG2 away from its binding interface with the foot domain of subunit C (C_{foot}) (see Appendix Fig. S4A). The cryoEM structures of V₁(C)Oxr1p (7fde) and autoinhibited V₁ (7tmm) likely capture stable, low energy conformations of the complexes. More structural work is needed to visualize additional structures on the conformational landscape during enzyme disassembly. We have included the following text in the Discussion to address the reviewer's question:

Page 20:

*"The binding site between Oxr1p and state 1 V_1V_o or subunit C containing $V_1\Delta H$ ($V_1(C)Oxr1p$) is formed in part by the C-terminal domain of the B subunit that constitutes the open catalytic site (**Appendix Figure S4D**). Upon conversion to state 2 via ATP hydrolysis, the B subunit C-terminal domain moves away from the Oxr1p binding site to bind the C-terminus of the neighboring A subunit to close the catalytic site, a conformational change that likely lowers the affinity for and facilitates release of Oxr1p (and with it, subunit C)."*

Referee #2:

-Does pRS316 belong to high-copy or low-copy number plasmid? What's the plasmid ORI? If pRS316 is a high-copy-number plasmid, the Oxr1NP is also over-expressed compared to wide-type of Oxr1. The authors may want to use other low-copy-number plasmid to repeat the experiments.

Authors' response:

A low copy plasmid pRS316 [CEN6, URA3] was used in the experiment. This has now been clarified in the main text:

Page 5:

"We then transformed $oxr1\Delta$ with a low-copy number plasmid (pRS316, CEN6, URA3) for expression of C-terminally HA-tagged Oxr1p (Oxr1p-HA) under its native promoter (strain $oxr1^{NP}$), or with a high-copy number plasmid (YEp352, 2 μ , URA3) for overexpression of Oxr1p-HA (strain $oxr1^{OE}$)."

Referee #2:

-For Fig 2C, whereas ATPase activities of Oxr1 Δ and wild type vacuoles were similar, those from Oxr1NP strain was reduced to ~75% of wild type. However, For Fig 2D, the relative V_1 to V_o ratio for Oxr1NP has no significant difference to wide type, whereas Oxr1 Δ was significantly increased. Why?

Authors' response:

The $oxr1^{NP}$ strain was used as a control for the endogenous level of *OXR1* expression via the pRS316 vector. pRS316 is considered a single or low-copy number plasmid (see above), meaning that some cells may have had more than one copy of the plasmid (see refs. 1 & 2 below). As result, one possibility for the lower ATPase activity of purified vacuoles from the $oxr1^{NP}$ strain could be due to slight overexpression of *OXR1* in some of the cells. However, why the relative ratio of V_1 to V_o is not significantly lower in the $oxr1^{NP}$ strain is not clear at this point.

On the other hand, while the relative ratio of V_1 to V_o is higher in the $oxr1\Delta$ strain, activity is similar to wild type. As discussed in the Result and Discussion sections of the manuscript, the higher $V_1:V_o$ ratio is likely due to the lack of Oxr1p mediated enzyme disassembly in the $oxr1\Delta$ strain (reversible disassembly is not an all or none process, even in glucose fed condition, ~30% of the enzyme remains disassembled, see ref. 3). However, as the activity in the $oxr1\Delta$ background remains the same as in WT, this suggests that there is a population of holoenzymes that contain V_1 subcomplexes that are inactive due to e.g. oxidative damage (see refs. 4, 5 & 6), and thus do not contribute to the activity, while significantly increasing the ratio of V_1 to V_o in the $oxr1\Delta$ strain. The presence of some inactive enzyme on the vacuole is also supported by our experiments with vacuoles and Oxr1p where we observed that while activity is reduced close to ConA inhibition levels, a significant amount of V_1 remains on the vacuole (Khan et al, 2022). This has been clarified in the main text:

Page 17:

*"Of note, whereas the *OXR1* deletion resulted in ~40% more V_1 on the vacuole, the ATPase activity remained similar to the wild type control. Earlier work had shown that even wild type vacuoles contain ~30% inactive (oxidized) V-ATPases {Oluwatosin, 1997 #54}, and it is possible that even more inactive complexes accumulate*

on vacuoles in absence of *Oxr1p*. This could explain why the higher $V_1:V_0$ ratio on *oxr1Δ* vacuoles does not result in higher activity, supporting a role for *Oxr1p* in quality control."

References:

- (1) Zhou, P., Szczypka, M. S., Young, R., & Thiele, D. J. (1994). A system for gene cloning and manipulation in the yeast *Candida glabrata*. *Gene*, 142(1), 135-140.
- (2) Sikorski, R. S., & Hieter, P. (1989). A system of shuttle vectors and yeast host strains designed for efficient manipulation of DNA in *Saccharomyces cerevisiae*. *Genetics*, 122(1), 19-27.
- (3) Kane, P. M. (1995). Disassembly and reassembly of the yeast vacuolar H⁺-ATPase in vivo. *Journal of Biological Chemistry*, 270(28), 17025-17032.
- (4) Feng, Y., & Forgac, M. (1992). A novel mechanism for regulation of vacuolar acidification. *Journal of Biological Chemistry*, 267(28), 19769-19772.
- (5) Liu, Q., Leng, X. H., Newman, P. R., Vasilyeva, E., Kane, P. M., & Forgac, M. (1997). Site-directed mutagenesis of the yeast V-ATPase A subunit. *Journal of Biological Chemistry*, 272(18), 11750-11756.
- (6) Oluwatosin, Y. E., & Kane, P. M. (1997). Mutations in the *CYS4* gene provide evidence for regulation of the yeast vacuolar H⁺-ATPase by oxidation and reduction in vivo. *Journal of Biological Chemistry*, 272(44), 28149-28157.

Referee #2:

-For Fig 2E, the error bar is too large. Please repeat more times or recalculate the band gray value because each experimental group has one outlier spot, respectively.

Authors' response:

The experiment has been repeated two more times to reduce the size of the error bar. See the revised figure panel below.

Fig. 2E (bottom panel): Averages of relative band intensities from five experiments. Data are presented as mean \pm SEM.

Referee #2:

-What's the Pearson's co-efficient for Fig 4A?

Authors' response:

We now have included the Pearson's coefficient for Fig 4A. See also the response to Reviewer 1 above.

Referee #2:

-For Fig 5B, indicate the incubation time for Oxr1, MgATP, MgATP+Oxr1.

Authors' response:

As mentioned in the legend of the figure, the incubation time of the experiment in Fig. 5B was 30 min (Legend of Fig 5B: "*ATPase activities of purified V-ATPase pre-incubated for **30 min** with a 2-fold molar excess of Oxr1p in absence (cyan) or presence (magenta) of 4 mM MgATP. As a control, ATPase activity was measured in presence of 4 mM MgATP only (green)*").

This now has been clarified in the main text as well:

Page 9-10:

*"We therefore asked whether ATP might affect the rate of Oxr1p mediated disassembly. To address this question, we incubated wild type V-ATPase with a two-fold molar excess of Oxr1p or ATP individually or in combination for **30 min**, and measured ATPase activities. Whereas Oxr1p alone resulted in ~30% loss of activity, inhibition was close to complete in presence of Oxr1p and ATP (Fig 5B), with negative stain EM confirming virtually complete V-ATPase disassembly (Fig 5C and D)."*

Referee #2:

-For Fig 6D, indicate the K_d for $V_1\Delta H(\Delta Oxr1)$. Compare $V_1\Delta H(\Delta Oxr1)$ with $V_1\Delta H(\Delta Oxr1)+C$, and analysis the binding affinity in the corresponding context.

Authors' response:

We now have provided the K_d for $V_1\Delta H(\Delta Oxr1)$ binding to bio-Oxr1p, see Results section and legend to Fig. 6D:

Page 13:

*"From the BLI experiments, we find that bio-Oxr1p binds $V_1\Delta H(\Delta Oxr1)$ both in absence and presence of subunit C, with **K_d s of $\sim 67 \pm 33$ nM and $\sim 7 \pm 1$ nM**, respectively (Fig 6D), consistent with our earlier ATPase inhibition assays and the stability of the $V_1(C)Oxr1p$ complex (Khan et al, 2022)."*

Legend of Fig 6D:

*"BLI analysis of the interaction between immobilized bio-Oxr1p and $V_1\Delta H$ purified from the $V_1\Delta H(\Delta Oxr1)$ strain in absence (left) and presence (right) of subunit C. Global fitting of the BLI traces obtained in presence of C gave a **K_d of $\sim 7 \pm 1$ nM** (right panel). Note that in absence of C (left panel), global fitting was less reliable due to the relatively low BLI signal combined with significant non-specific binding of $V_1\Delta H$ to empty sensors, with a resulting estimate of the **K_d of $\sim 67 \pm 33$ nM**."*

Referee #2:

-In page 3, Oxr1p should stand for Oxidation Resistance protein 1. Please unify all the usage of Oxr1 and Oxr1p throughout this manuscript.

Authors' response:

We thank the reviewer for bringing that to our attention. We have now changed the nomenclature of protein and gene names to more closely align with the nomenclature as given in e.g. the Saccharomyces Genome Database (SGD). We now use *OXR1* when referring to the gene, *oxr1 Δ* when referring to the yeast strain in which *OXR1* is deleted from the genome, and Oxr1p when referring to the Oxr1 protein.

Referee #2:

-Because the structures in Fig 1 is not new results, suggest authors change it to a Expanded View figure and reorder the sequences of all the figures.

Authors' response:

We appreciate the Reviewer's suggestion to move Fig. 1 to Fig. EV1. However, we think that keeping Fig. 1 as part of the Introduction will help readers to quickly grasp the premise of this study, particularly when reading the printed version of the article, where EV figures are not placed within the main text.

Referee #2:

-For Discussion, it not that accurately interpret the results, but repetitive with the Results section and it even contain a comprehensive review of the field to some degree. Please rephrase it, especially compact the content related to Figure 8.

Authors' response:

We thank the reviewer for the suggestion. We revised parts of the Discussion and reduced its length from the original 2573 to ~2190 words in the revised manuscript. We also specifically shortened the content related to the discussion of the overall model in Fig. 8. See the version of the revised manuscript with tracked changes.

Dear Stephan,

Thank you for the submission of your revised manuscript to EMBO Reports. Please apologize my delayed response but I have meanwhile checked your response to the referee concerns and their requests for clarification and I have also checked your manuscript from the editorial side.

I am now writing with an 'accept in principle' decision, which means that I will be happy to accept your manuscript for publication once a few minor issues/corrections have been addressed, as follows.

- I note that the structure shown in Figure 1 has been published before, as also noted by referee #2. You indicated that these panels are part of the introduction to help readers grasp the premise of the study. I agree that these panels will be helpful, but also ask you to reconsider whether showing the actual structure in panel B is really necessary or whether it could be replaced by a schematic, akin to what is shown in panel A.

- Please update the 'Conflict of interest' paragraph to our new 'Disclosure and competing interests statement'. For more information see

<https://www.embopress.org/page/journal/14693178/authorguide#conflictsofinterest>

- Please remove the Author Contributions from the manuscript file and make sure that the author contributions in our online submission system are correct and up-to-date. The information you specified in the system will be automatically retrieved and typeset into the article. You can enter additional information in the free text box provided, if you wish.

- Gordon DE et al, 2020 refers to a preprint. Please formally cite this preprint in the text and in the reference list as follows: The citation in the text is: (preprint: NAME1 et al, YEAR); in the reference list: Author NAME1, Author NAME2 (YEAR) article title. bioRxiv doi [PREPRINT].

- Data citations: We encourage to include datasets obtained from public resources in the reference list in the form of data citations (e.g, when referring to deposited structures). The original publication (article) that reported the respective dataset should additionally be included as a regular reference.

Data references should include authors, when possible, the year, the full name of the database where the data is available, the accession number or DOI and, importantly, a resolvable link that points directly to the dataset. The resolvable link can be in the form of either a plain URL, a DOI or an identifiers.org construct. Each data reference should be labeled with the tag "[DATASET]" at the end to distinguish it from the other references.

Example:

References

Hörnberg E, Ylitalo EB, Crnalic S, Antti H, Stattin P, Widmark A, Bergh A, Wikström P (2011) Gene Expression Omnibus GSE29650 (<https://www.ncbi.nlm.nih.gov/geo/query/acc.cgi?acc=GSE29650>). [DATASET]

Hörnberg E, Ylitalo EB, Crnalic S, Antti H, Stattin P, Widmark A, Bergh A, Wikström P (2011) Expression of androgen receptor splice variants in prostate cancer bone metastases is associated with castration-resistance and short survival. PLoS One 6: e19059

In the main text, these datasets should be cited with the prefix "Data ref:" to distinguish them from the reference to the original article that reported the dataset. Example:

"...were grouped based on the relative levels of AR-Vs expressed, mainly AR-V7 (Hörnberg et al, 2011; Data ref: Hörnberg et al, 2011)."

- Appendix: please add page numbers to the table of content.

- Movie EV1: the legend needs to be removed from the manuscript file and provided in a readme.txt file; both the movie and the legend should be zipped together and the folder should be uploaded as Movie EV1.

- Data availability section: Please only keep the text that refers to the datasets submitted in external repositories and remove the first two sentences. Please insert links that resolve directly to the cryoEM maps deposited in EMD.

- On page 28 you mention a previously generated but unpublished construct. Please make sure to provide all information necessary to describe the cloning and generation of the construct. I think it is not necessary to call it 'unpublished' since you are in effect publishing and describing it in this study.

- Our production/data editors have asked you to clarify several points in the figure legends (see below). Please incorporate these changes in the manuscript and return the revised file with tracked changes with your final manuscript submission.

- a) Please note that a separate 'Data Information' section is required in the legends of figures 6b-c, f. The Data Information section provides information that is valid for all or several panels.
- b) Please note that in figures 3a-c; 4b-c; 5a-b; EV 2a; there is a mismatch between the annotated p values in the figure legend and the annotated p values in the figure file that should be corrected, i.e., only the p-values that are actually shown in the figure panels should be defined in the figure legends.

- On a different note, I would like to alert you that EMBO Press offers a new format for a video-synopsis of work published with us, which essentially is a short, author-generated film explaining the core findings in hand drawings, and, as we believe, can be very useful to increase visibility of the work. This has proven to offer a nice opportunity for exposure i.p. for the first author(s) of the study. Please see the following link for representative examples and their integration into the article web page:

<https://www.embopress.org/doi/full/10.15252/emj.2019103932>

I am looking forward to receiving the final version of your manuscript.

With kind regards,

Martina

All editorial and formatting issues were resolved by the authors.

Dr. Stephan Wilkens
SUNY Upstate Medical University
Biochemistry and Molecular Biology
4237 Weiskotten Hall
750 East Adams Street
Syracuse, NY 13210
United States

Dear Stephan,

I am very pleased to accept your manuscript for publication in the next available issue of EMBO reports. Thank you for your contribution to our journal.

Best regards,

Martina
